# Distributed Lion for Communication Efficient Distributed Training

**Bo Liu**[*]
The University of Texas at Austin
bliu@cs.utexas.edu

**Lemeng Wu**[*]
Meta AI
lmwu@google.com

**Lizhang Chen**[*]
The University of Texas at Austin
lzchen@utexas.edu

**Kaizhao Liang**
The University of Texas at Austin
kaizhaol@utexas.edu

**Jiaxu Zhu**
Meta AI
jiaxuzhu@meta.com

**Chen Liang**
Google
crazydonkey@google.com

**Raghuraman Krishnamoorthi**
Meta AI
raghuraman@meta.com

**Qiang Liu**
The University of Texas at Austin
lqiang@cs.utexas.edu

## Abstract

The Lion optimizer has been a promising competitor with the AdamW for training large AI models, with advantages in memory, computation, and sample efficiency. In this paper, we introduce Distributed Lion, an innovative adaptation of Lion for distributed training environments. Leveraging the sign operator in Lion, our Distributed Lion only requires to communicate binary or lower-precision vectors between workers to the center server, significantly reducing the communication cost. Our theoretical analysis confirms Distributed Lion's convergence properties. Empirical results demonstrate its robustness across a range of tasks, worker counts, and batch sizes, on both vision and language problems. Notably, Distributed Lion attains comparable performance to standard Lion or AdamW optimizers applied on aggregated gradients, but with significantly reduced communication bandwidth. This feature is particularly advantageous for training large models. In addition, we also demonstrate that Distributed Lion presents a more favorable performance-bandwidth balance compared to existing efficient distributed methods such as deep gradient compression and ternary gradients.

## 1 Introduction

The pursuit of modern artificial intelligence hinges on the training of large-scale models like large language models[28] and large vision models (LVM)[20]. As the stakes – in terms of time, cost, and environmental impact – grow ever higher for training expansive AI systems, the hunt for efficient optimizers becomes critical.

Recently, a new optimization named Lion (evolved sign momentum) [11] has been discovered with an evolutionary program. It was shown that it exhibits performance on par with the current state-of-the-art AdamW [26] across a wide range of tasks, while reducing the memory cost and training time.

---

[*]Equal contribution.

38th Conference on Neural Information Processing Systems (NeurIPS 2024).

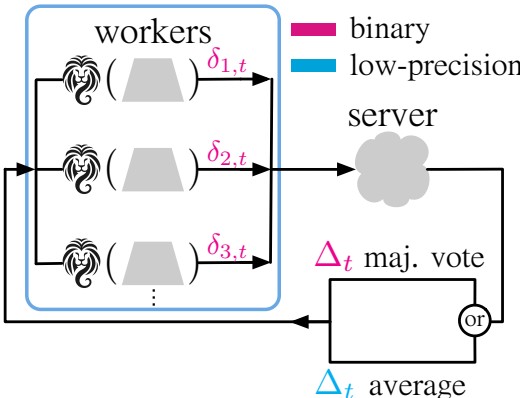

Figure 1: Illustration of Distributed-Lion. Each worker keeps its *own* optimizer state and applies the Lion optimizer individually to a binary update $\delta_{i,t} = \text{Lion}(x, \mathcal{D}_i)$ (without the weight decay), then the server aggregates all $\delta_{i,t}$ to produce a binary $\Delta_t$ by majority vote (or an integer $\Delta_t$ by averaging) and send it back to all workers. The workers then apply $\Delta_t$ and weight decay to update their model parameters (Algorithm 1).

Consider optimizing a loss function $f_{\mathcal{D}}(x)$ on $\mathbb{R}^d$ with a dataset $\mathcal{D}$, the update rule of Lion is:

$$m_{t+1} = \beta_2 m_t + (1 - \beta_2)\nabla f_{\mathcal{D}}(x_t),$$
$$\delta_t = \text{Lion}(x_t, \mathcal{D}) \overset{def}{=} \text{sign}(\beta_1 m_t + (1 - \beta_1)\nabla f_{\mathcal{D}}(x_t)), \tag{1}$$
$$x_{t+1} = x_t - \epsilon(\delta_t + \lambda x_t),$$

where $m_t$ plays the role of the momentum, $\epsilon$ is the learning rate, $\beta_1, \beta_2 \in [0, 1]^2$ are two momentum related coefficients, and $\lambda \geq 0$ is the weight decay coefficient. Comparing Lion against AdamW, one observes that Lion only requires the storage of the first-order momentum term, which results in a more relaxed memory requirement.

In this study, we tailor the Lion optimizer for distributed training. The Lion optimizer is particularly suitable for this context due to two main attributes: (1) its simple update mechanism that relies solely on first-order momentum, and (2) its use of the $\text{sign}(\cdot)$ function. We showcase the effective employment of the $\text{sign}(\cdot)$ function to streamline communication processes, leading to the development of a novel distributed training framework named Distributed Lion. Within the Distributed Lion framework, each participating worker independently adjusts the model parameters using a distinct instance of the Lion optimizer, thereby maintaining separate optimizer states. A distinctive feature of this framework is the mode of communication between workers and the central server, which is restricted to binary or low-precision vectors.

Crucially, in this setup, workers convey updates rather than raw gradients to the central server. The server, in turn, aggregates these updates through either a straightforward averaging process (Distributed Lion-Avg) or a majority voting mechanism (Distributed Lion-MaVo). In the case of Distributed Lion-MaVo, the consolidated update is maintained as a binary vector, whereas for Distributed Lion-Avg, given the presence of $n$ workers, each element of the update vector is encoded using $\log(n)$ bits. This approach markedly reduces the bandwidth requirements compared to traditional distributed training methods, which typically rely on high-precision floating-point vectors for communication. The bandwidth efficiencies achieved by our method are detailed in Table 1. Our contributions are: **1)** We introduce the Distributed Lion algorithm, a simple yet effective approach to extend Lion to distributed training, where all communications between workers and the server are done through binary or low-precision vectors (Section 2); **2)** We provide theoretical analysis to ensure the convergence of Distributed Lion (Section 3); **3)** Empirically, we demonstrate that on both vision and language modeling tasks, Distributed Lion achieves comparable performance against applying Lion and Adam with the synchronized gradients from all workers, while being significantly more communication efficient. In addition, we show that Distributed Lion achieves a better trade-off than existing efficient distributed training methods like deep gradient compression [24] and ternary gradients [36] (Section 5).

---

[2]Chen et al. [11] suggests ($\beta_1 = 0.9, \beta_2 = 0.99$) based on empirical findings.

| Method | Bandwidth Requirement | |
|---|---|---|
| | Worker→Server | Server→Worker |
| Global Lion/AdamW | $32d$ | $32d$ |
| TernGrad [36] | $1.5d$ | $\log(2n+1)d$ |
| DGC [24] | $(1-\eta)32d$ | $32d$ |
| Distributed Lion-Avg | $d$ | $\log(n)d$ |
| Distributed Lion-MaVo | $d$ | $d$ |

Table 1: Minimum bandwidth requirements of different methods for a model with $d$ parameters and $n$ workers. For Deep Gradient Compression (DGC), $\eta$ denotes the compression rate (default: $\eta = 0.96$).

## 2 The Distributed Lion

We introduce the distributed learning problem and then our Distributed Lion framework.

### 2.1 Distributed Training

In distributed training, we aim to minimize the following learning objective:

$$\min_x F(x) = \frac{1}{N} \sum_{i=1}^{N} \mathbb{E}_{\xi_i \sim \mathcal{D}_i} \left[ f(x; \xi_i) \right]. \tag{2}$$

Here, $N$ denotes the number of workers, $\{\mathcal{D}_i\}$ are $N$ datasets,[3] and $x$ is the model parameter (e.g., the weights of a neural network). In the distributed learning setting, each worker $i \in [n]$ will get its own dataset $\mathcal{D}_i$, and we assume there is a centralized server that all workers can communicate with. The simplest distributed training technique is to perform distributed gradient aggregation:

$$g_{\text{server}} = \frac{1}{N} \sum_{i=1}^{N} g_i, \quad \text{where} \quad g_i = \mathbb{E}_{\xi_i \sim \mathcal{D}_i} \left[ \nabla_x f(x; \xi_i) \right]. \tag{3}$$

Here, each local gradient $g_i$ is an unbiased estimation of the true gradient $\nabla_x F(x)$ when $\mathcal{D}_i$ are i.i.d. drawn from the same underlying distribution. The server aggregates all local gradients into $g_{\text{server}}$, and then applies an optimizer like Adam [19] on top of $g_{\text{server}}$. However, the aggregation step requires communicating the full gradient vectors $g_i$, which can be expensive for large models.

**Notation.** Given a function $f(x; \xi)$, the gradient $\nabla f(x; \xi)$ is taken with respect to variable $x$. We use $\|\cdot\|$, $\|\cdot\|_1$, and $\|\cdot\|_\infty$ to denote the $\ell_2$, $\ell_1$, and $\ell_\infty$ norm, respectively. $\xi_{i,t}$ is the sampled data at time $t$ for the $i$-th worker and $g_{i,t} = \nabla f(x_t; , \xi_{i,t})$. We similarly denote $z_{i,t}$ as any variable $z$ at time $t$ from worker $i$.

### 2.2 Distributed Lion

The main idea of Distributed Lion is to leverage the binary nature of the Lion's update for efficient communication. To enable that, we want the workers to *only send the binary updates* to the server. As a result, we let each worker keep tracks of its own optimizer state, i.e., the momentum $m_{i,t}$. Then at each step, each worker $i$ first computes:

$$\begin{aligned} m_{i,t+1} &= \beta_2 m_{i,t} + (1-\beta_2) g_{i,t}, \\ \delta_{i,t} &= \text{sign}(\beta_1 m_{i,t} + (1-\beta_1) g_{i,t}). \end{aligned} \tag{4}$$

Then all workers send the $\delta_{i,t}$ back to the server. The server receives the binary "updates" from all workers and then aggregates them. Here, we propose two simple ways for aggregation. Denote $S_t = \sum_{i=1}^{N} \delta_{i,t}$, which is a vector of integers in $\{0, \ldots N\}$. Define the aggregation as follows:

$$\Delta_t = \text{aggregate}(S_t) = \begin{cases} \frac{1}{N} S_t & \text{(Averaging)} \\ \text{sign}(S_t) & \text{(Majority Vote)} \end{cases}. \tag{5}$$

---

[3]Throughout this work, we assume $\{\mathcal{D}_i\}$ consist of i.i.d data samples, $\xi_i$ sampled from $\mathcal{D}_i$ is i.i.d. though our method should be directly applicable to non-i.i.d data.

---

**Algorithm 1** Distributed Lion Training

---

**Inputs:** Initial parameters $x_0 \in \mathbb{R}^d$, datasets $\{\mathcal{D}_1, \ldots, \mathcal{D}_N\}$, loss function $f$, learning rate $\epsilon$, hyper-parameters $\beta_1, \beta_2 \in [0, 1]$ (default to $0.9, 0.99$)[4], and the weight decay $\lambda$.

**Initialization:** $t = 0$, $\forall i, m_{i,0} = \mathbf{0}$, and $x_{i,0} = x_0$.
**while** not convergent **do**
    **Worker-side:** Each worker $i$ samples a batch $\xi_{i,t} \in D_i$, computes the following, and sends $\delta_{i,t}$ to the server:

$$\text{if } t > 0, \ x_{i,t} \leftarrow x_{i,t-1} - \epsilon\big(\Delta_{t-1} + \lambda x_{i,t-1}\big)$$
$$\delta_{i,t} \leftarrow \text{sign}\big(\beta_1 m_{i,t} + (1 - \beta_1)\nabla_x f(x_{i,t}; \xi_{i,t})\big)$$
$$m_{i,t+1} \leftarrow \beta_2 m_{i,t} + (1 - \beta_2)\nabla_x f(x_{i,t}; \xi_{i,t}).$$

    **Server-side:** The server computes the aggregated update $\Delta_t$ and broadcast it to all workers:

$$\Delta_t = \begin{cases} \frac{1}{N}\big(\sum_{i=1}^N \delta_{i,t}\big) & \text{(Averaging)} \\ \text{sign}\big(\sum_{i=1}^N \delta_{i,t}\big) & \text{(Majority Vote)} \end{cases} \quad \text{and} \quad t \leftarrow t + 1.$$

**end while**

---

So we simply average or take the majority vote from all $\{\delta_{i,t}\}$. Here, we denote binary vectors in magenta and low precision vectors in cyan. In the end, the server broadcasts $\Delta_t$ back to each worker $i$, and each worker performs $x_{i,t+1} = x_{i,t} - \epsilon(\Delta_t + \lambda x_{i,t})$, where $\epsilon$ is the step size and $\lambda$ is the weight decay coefficient.

**Communication Cost** In both variants of Distributed Lion, the $N$ workers only need to send the binary vectors $\delta_{i,t}$ to the server. The server then sends the aggregated update $\Delta_t$ back to the workers, which is binary when using the majority vote aggregation, and an integer in $\{0, \ldots, N\}$ when using the averaging aggregation. Note that an integer in $\{0, \ldots, N\}$ can be represented by at most $\log(N)$ bits. In practice, usually $N \ll 2^{32}$ hence $\log(N) < 32$ and we still save the communication bandwidth even with the average aggregation, comparing against communicating with floating point numbers (Check Table 1). The full Distributed Lion algorithm is summarized in Algorithm 1.

## 3 Theoretical Analysis

We provide our theoretical analysis of the Distributed Lion algorithm, both with the averaging and the majority vote aggregation methods. In the following, we first describe that the distributed training problem can be viewed as a constrained optimization problem when Distributed Lion is used. We provide convergence results for Distributed Lion with both aggregation methods.

### 3.1 Lion as Constrained Optimization

Chen et al. [10] showed that the (global) Lion is a theoretically novel and principled approach for minimizing a general loss function $f(x)$ while enforcing a box-constrained optimization problem:

$$\min_{x \in \mathbb{R}^d} f(x) \quad s.t. \quad \|\lambda x\|_\infty \leq 1, \tag{6}$$

where the constraint is introduced due to the use of the weight decay coefficient $\lambda$. Moreover, Chen et al. [10] showed that the Lion dynamics consists of two phases:

1) **[Phase 1]** When the constraint is not satisfied, that is, $x \notin \mathcal{F}$, where $\mathcal{F}$ is the feasible set $\mathcal{F} \overset{def}{=} \{x \colon \|\lambda x\|_\infty \leq 1\}$, it exponentially decays the distance to $\mathcal{F}$: $\exists \, \alpha \in (0, 1)$, such that
$$\text{dist}(x_{t+n}, \mathcal{F}) \leq \alpha^n \text{dist}(x_t, \mathcal{F}).$$
where $n \geq 0$. Hence, $x_t$ converges to $\mathcal{F}$ rapidly and stays within $\mathcal{F}$ once it reaches it.

2) **[Phase 2]** After $\lambda x_t$ enters $\mathcal{F}$, the dynamics minimizes the objective $f(x)$ while being confined within the set $\mathcal{F}$. This step is proved in [10] by constructing a Lyapunov function when $\text{sign}(\cdot)$ is treated as the sub-gradient of a convex function.

## 3.2 Convergence Analysis

In this section, we analyze the convergence of distributed Lion algorithms. Similar to the case of global Lion, we show that distributed Lion also solves the box constrained optimization (6). Its dynamics also unfolds into two phases aligning with Lion's dynamics: Phase I shows rapid convergence to a feasible set $\mathcal{F}$, while Phase II seeks to minize the objective $f(x)$ within the feasible set $\mathcal{F}$. Different from the Lyapunov approach used in Chen et al. [10], the proof of our Phase II result is made by introducing a surrogate metric $\mathcal{S}(x)$ of constrained optimality, and providing upper bound of $\mathcal{S}(x_t)$ following the algorithm. Our analysis makes the following assumptions.

**Assumption 3.1** (Variance bound). *$\mathcal{D}_i$ is i.i.d. drawn from a common distribution $\pi_*$, and the stochastic sample $\xi^i \sim \mathcal{D}_i$ is i.i.d. and upon receiving query $x \in \mathbb{R}^d$, the stochastic gradient oracle gives us an* independent *unbiased estimate $\nabla f(x; \xi^i)$ from the $i$-th worker that has coordinate bounded variance:*

$$\mathbb{E}_{\xi}[\nabla f(x; \xi^i)] = \nabla f(x), \quad \mathbb{E}_{\xi}\left[\|\nabla f(x; \xi^i) - \nabla f(x)\|^2\right] \leq \sigma^2.$$

**Assumption 3.2** (Smooth and Differentiable $f$). *Function $f(\cdot)$ is differentiable and $L$-smooth.*

**Assumption 3.3** (Bias Correction). *Consider the sequence $\{m_t^i\}_{t>0, i\in[N]}$ generated by Algorithm 1, $\mathbb{E}[\tilde{m}_t^i]/\mathbb{E}[\mathrm{sign}(\tilde{m}_t^i)] \geq 0$.*

Note that assumption 3.1 and 3.2 are standard in the analysis of stochastic optimization algorithms [8, 34]. When Assumption 3.1 holds, $\mathbb{E}\|\frac{1}{N}\sum_{i=1}^{N}\nabla f(x; \xi_i) - \nabla f(x)\|^2 \leq \sigma^2/N$. In distributed training setting, $m_{1,t}, m_{2,t}, \cdots, m_{N,t}$ are i.i.d., so $\mathbb{E}[\beta_1 m_{i,t} + (1-\beta_1)g_{i,t}]$ and $\mathbb{E}[\mathrm{sign}(\tilde{m}_{t+1}^i)]$ don't depend on $i$. Assumption 3.3 evaluates the discrepancy between the expected value and the expected sign of a measure, positing that the expected values of $\tilde{m}_t^i$ and $\widetilde{\mathrm{sign}}(m_t^i)$ ought to share the same sign.

We now present our results. Similar to the case of global Lion, the dynamics of distributed lion can also be divided into two phases depending on if the constraint $x \in \mathcal{F}$ is satisfied.

**Phase I** ($x \notin \mathcal{F}$)   In line with the behavior observed in the global Lion, when the constraint is not satisfied, both variants of distributed Lion decrease the distance to the feasible set exponentially fast.

**Theorem 3.4** (Phase I). *Assume $f: \mathbb{R}^d \to \mathbb{R}$ is $L$-smooth, $\beta_1, \beta_2 \in (0,1)$, and $\beta_2 > \beta_1$, and $\epsilon, \lambda > 0$. Let $(x_t)_{t\geq 0}$ be generated by Algorithm 1. Define $\mathcal{F} = \{x: \|\lambda x\|_{\infty} \leq 1\}$, and $\mathrm{dist}(x_t, \mathcal{F}) = \inf_{z\in\mathcal{F}}\|z - x_t\|$ w.r.t. any norm $\|\cdot\|$. For any two non-negative integers $s \leq t$, then $\forall s \leq t$, we have*

$$\mathrm{dist}(x_t, \mathcal{F}) \leq (1 - \epsilon\lambda)^{t-s}\mathrm{dist}(x_s, \mathcal{F}).$$

Hence, $x_t$ converges to $\mathcal{F}$ rapidly and stays within $\mathcal{F}$ once it arrived.

**Phase II** ($x \in \mathcal{F}$)   Now, we present the main result of the analysis for Phase II in Theorems 3.6, 3.7, and 3.8. We start with introducing a surrogate metric that quantifies the optimality of the solution within Phase II:

$$\mathcal{S}(x) := \langle \nabla f(x), \mathrm{sign}(\nabla f(x)) + \lambda x \rangle. \tag{7}$$

Let's delve into the implications of $\mathcal{S}(x) = 0$.

**Proposition 3.5.** *Assume $f$ is continuously differentiable, $\lambda > 0$, and $\|\lambda x\|_{\infty} \leq 1$. Then $\mathcal{S}(x) = 0$ implies a KKT stationary condition of $\min_x f(x)$ s.t. $\|\lambda x\|_{\infty} \leq 1$.*

This KKT score (7) is tailored to encompass the stationary solutions of the box constrained problem as described in (6). Building on this, we then proceed to analyze the convergence for the majority vote, averaging, and global LION strategies throughout this section.

**Theorem 3.6** (Majority Vote). *Assumptions 3.1, 3.2, and 3.3 hold, consider the Majority vote scheme in Algorithm 1 , $\beta_1, \beta_2 \in (0,1)$, and $\beta_2 > \beta_1$, and $\sigma \leq 2\sqrt{d}\beta_1\beta_2^t\|\nabla f(x_0)\|, 1 \leq t \leq T$ , and $\epsilon, \lambda > 0$. Let $(x_t)_{t\geq 0}$ be generated by Majority Vote, and it is in Phase II: $\|\lambda x_t\|_{\infty} \leq 1$ for all $t$.*

*We have*

$$\frac{1}{T}\sum_{t=1}^{T}\mathbb{E}[\mathcal{S}(x_t)] \leq \frac{f(x_0) - f^*}{T\epsilon} + \frac{2D\beta_1\beta_2\sqrt{d}\|\nabla f(x_0)\|}{T(1-\beta_2)} + \frac{4\beta_1 L\epsilon d}{1-\beta_2} + \frac{2\sqrt{d}\sigma(1+\sqrt{C}) + 2\rho}{\sqrt{N}} + 2L\epsilon d,$$

$$\tag{8}$$

*where $C = \beta_1^2(1-\beta_2)\frac{1}{1+\beta_2} + (1-\beta_1)^2$, and $D = \max\{1, \sigma/(2\sqrt{d}\beta_1\beta_2^T\|\nabla f(x_0)\|)\}$,*

$$\rho_t[k] = \begin{cases} 0 & \text{if } \mathbb{E}[\text{sign}(\tilde{m}_{t+1}^i[k])] = 0, \\ \mathbb{E}[\tilde{m}_{t+1}^i[k]]/\mathbb{E}[\text{sign}(\tilde{m}_{t+1}^i[k])] & \text{otherwise} \end{cases}$$

*, and $\rho = \max_{1 \le t \le T} \|\rho_t\|$.*

The result above shows that $\frac{1}{T}\sum_{t=1}^T \mathbb{E}[\mathcal{S}(x_t)]$ decays with a rate of $\mathcal{O}(\frac{1}{T\epsilon} + \frac{1}{T(1-\beta_2)} + \epsilon + \frac{1}{\sqrt{N}})$. This rate is in fact on par with global Lion as we show in the following result.

**Theorem 3.7** (Global). *Assumptions 3.1 and 3.2 hold, Consider the scheme in Algorithm (16), with the same settings in Theorem 3.6, we have*

$$\frac{1}{T}\sum_{t=1}^T \mathbb{E}[\mathcal{S}(x_t)] \le \frac{f(x_0) - f^*}{T\epsilon} + \frac{2\beta_1\beta_2\sqrt{d}\|\nabla f(x_0)\|}{T(1-\beta_2)} + \frac{4\beta_1 L\epsilon d}{1-\beta_2} + \frac{2(1-\beta_1)\sqrt{d}\sigma}{\sqrt{N}} + 2L\epsilon d. \quad (9)$$

**Theorem 3.8** (Averaging). *Assumptions 3.1 and 3.2 hold, consider the Averaging scheme in Algorithm 1 , with the same settings in Theorem 3.6, we have*

$$\frac{1}{T}\sum_{t=1}^T \mathbb{E}[\mathcal{S}(x_t)] \le \frac{f(x_0) - f^*}{T\epsilon} + \frac{2\beta_1\beta_2\sqrt{d}\|\nabla f(x_0)\|}{T(1-\beta_2)} + \frac{4\beta_1 L\epsilon d}{1-\beta_2} + \frac{2\beta_1\sqrt{d}\sigma}{\sqrt{1+\beta_2}} + 2(1-\beta_1)\sqrt{d}\sigma + 2L\epsilon d$$

$$(10)$$

The Averaging method's convergence bound doesn't improve with more workers since $\frac{1}{N}\sum_{i=1}^N \text{sign}(\delta_{i,t})$ doesn't approximate $\text{sign}(\sum_{i=1}^N \delta_{i,t})$ effectively, unlike the Majority Vote's approach $\text{sign}(\sum_{i=1}^N \text{sign}(\delta_{i,t}))$.

# 4 Related Work

In this section, we provide a summary of optimizers that use the sign function and existing literature on bandwidth-friendly distributed training.

**Sign Operation in Optimization**    The sign operation is integral to optimization for several reasons. Primarily, it acts as a normalization mechanism by disregarding the magnitude of gradients, thereby equilibrating updates across different dimensions and potentially facilitating the avoidance of saddle points. Additionally, the binary nature of the sign function's output significantly reduces the memory footprint required for storing gradient updates. The concept of sign-based optimization dates back to RProp [30] and has seen renewed interest with the advent of SignSGD and its momentum-enhanced variant, Signum [4]. A more recent advancement is the generalized SignSGD algorithm introduced by [14], which incorporates a preconditioner, making it a superset of SignSGD and akin to Adam in certain aspects. A noteworthy addition to sign-based optimizers is the Lion optimizer, which emerged from evolutionary program search, achieving performance comparable to Adam [19] and AdamW [26] for the first time. Lion distinguishes itself from Signum by employing a different convex combination for outputting local updates, a technique referred to as the double-$\beta$ scheme, reminiscent of Nesterov's momentum update, and encapsulates Signum as a particular case. On the theoretical front, SignSGD and Signum have been shown to exhibit convergence rates comparable to traditional SGD [4]. Recent work by [34] has extended the theoretical understanding by providing a convergence theory that relaxes the requirements for bounded stochastic gradients and enlarged batch sizes. Additionally, Lion has demonstrated its capability in performing constrained optimization under the $\ell_\infty$-norm constraint [10].

**Distributed Training**    In addressing the communication constraints of distributed training, the research community has devised several innovative strategies, prominently featuring asynchronous Stochastic Gradient Descent (SGD), gradient quantization, and sparsification techniques. Asynchronous SGD offers a solution by enabling parameter updates immediately after back-propagation, bypassing the need for gradient synchronization, thereby expediting the training process [9, 40, 25]. Li et al. [21] utilizes sketch-based algorithms for lossless data compression [23], achieving an asymptotically optimal compression ratio [22]. However, its applicability is limited to highly sparse

gradients, making it orthogonal to our research. In the realm of gradient quantization, methods such as 1-bit SGD [33], QSGD [2], and TernGrad [36] are pivotal. These approaches compact the gradient data, substantially reducing the required communication bandwidth, with 1-bit SGD demonstrating a tenfold acceleration in speech applications and both QSGD and TernGrad confirming the feasibility of quantized training in maintaining convergence. Moreover, gradient sparsification further mitigates the communication load by transmitting only the most substantial gradients. Techniques like threshold quantization and Gradient Dropping [1] exemplify this, with Gradient Dropping notably achieving a 99 reduction in gradient exchange with minimal impact on performance metrics, such as a mere 0.3 loss in BLEU score for machine translation tasks. The recent Deep Gradient Compression (DGC) strategy [24] also contributes to this field by incorporating momentum correction and local gradient clipping among other methods to maintain accuracy while significantly reducing communication demands, albeit at the cost of increased computational overhead. Compared to gradient quantization methods, Distributed Lion uniquely leverages the binary nature of Lion's update and can be viewed as performing quantization on updates rather than the gradient.

## 5 Experiment

In this section, we perform a thorough evaluation of the Distributed Lion algorithm, employing both the averaging and majority vote aggregation methods. The design of our experiments is aimed at addressing the following questions to ascertain the algorithm's efficacy and performance:

**(Q1)** How does `Mavolion` perform in comparison to traditional global distributed training methods, which aggregate gradients from local workers to apply an optimizer to the collective gradient?

**(Q2)** How does `Mavolion` measure up against established methodologies known for their communication efficiency in distributed training?

**(Q3)** How does Distributed Lion scale on large vision or language problems?

### 5.1 Comparing Distributed Lion Against Established Methods on CIFAR-10

To address **Q1** and **Q2**, we compare Distributed Lion with both the averaging and the majority vote methods, against established low-bandwidth distributed training techniques and the global distributed training methods. We consider the following baseline methods: **1) Global AdamW (G-AdamW)**, where we apply AdamW with the averaged gradients from all workers. **2) Global Lion (G-Lion)**, where we apply Lion with the averaged gradients from all workers. Note that Global AdamW and Global Lion serve as the performance and communication upper bounds. **3) Distributed Lion with Averaged Updates (D-Lion (Avg))**, In contrast to the majority vote mechanism used in Distributed Lion, this variant averages the binary update vectors from all workers. While D-Lion (Avg) might offer improved performance in principle, it comes at the cost of non-binary communication from the server to the workers. **4) TernGrad** [36]. The main idea is to tenarize the gradient into a vector of $\{-1, 0, 1\}$, which is similar to what Lion does. But this process is done on the gradient level instead of on the update level **5) Gradient Dropping (GradDrop)** [1]. The main idea is to drop insignificant gradient entries and only transmit sparse gradient signals. **6) Deep Gradient Compression (DGC)** [24]. DGC is built on top of the GradDrop, but additionally applies momentum correction, local gradient clipping, momentum factor masking, and warm-up training.

**Experiment Setup**   For GradDrop, DGC, and TernGrad, we choose the compression rate of $0.04$ (note that $1/32 = 0.03125$) to match the bandwidth of the D-Lion (MaVo). We conduct experiments on the CIFAR-10 dataset using a vision transformer (ViT) with 6 layers, 8 heads, and a hidden dimension of 512. This is because ViT has arguably become the most widely used architecture in computer vision, and we empirically found no additional gain in performance when using a larger ViT on CIFAR-10. In addition, to validate how Distributed Lion performs with different numbers of workers, we consider $k \in \{4, 8, 16, 32\}$, each worker at each step samples an i.i.d batch of size 32.

We list the optimal hyperparameters selected for each method from Figure 2 in Table 4. The learning rates are selected from $\{0.00005, 0.001, 0.005, 0.01\}$ and the weight decays are selected from $\{0.0005, 0.001, 0.005\}$. For each experiment, we use a cosine learning rate scheduler and run for 200 epochs, and we ensure that in each epoch, each local worker sees the entire dataset once.

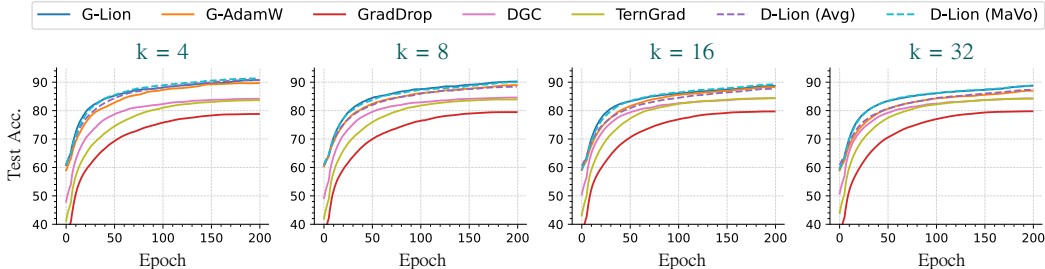

Figure 2: Performance of Distributed Lion v.s. baseline distributed optimizers on CIFAR-10 with 4, 8, 16, and 32 workers, each worker at each step runs on a local batch with size 32. All results are averaged over three seeds.

Each experiments are conducted with three random seeds $\{42, 52, 62\}$, which results in a total of $4 \times 7 \times 3 = 84$ experiments.

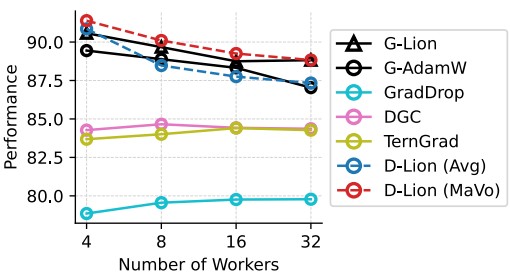

Figure 3: Performance of G-Lion, G-AdamW, Grad-Drop, DGC, TernGrad, and D-Lion (Avg/MaVo) v.s. the number of workers $k$.

Figure 4: Test Error v.s. Communication Bits per Iteration (closer to the lower-left is better). Note that we set G-Lion and G-AdamW are both 64, because they require 32 bits per parameter, and there are both worker-to-server and server-to-worker communications.

**Observation** We plot the testing accuracy (Test Acc.) over epochs for different methods in Figure 2, the best testing accuracy of different methods over the number of workers in Figure 3, and the performance versus per-iteration bandwidth in Figure 4 when using $k = 4$ workers. From the above plots, we make the following observations.

- Compared to global methods, D-Lion (MaVo) performs on par with G-Lion. D-Lion (Avg) performs slightly worse than G-Lion but is on par with G-Adamw (Figure 2).

- Compared to established communication efficient methods, both D-Lion (MaVo) and D-Lion (Avg) outperform GradDrop, DGC and TernGrad by a large margin (Figure 2).

- We observe that both D-Lion (MaVo) and D-Lion (Avg) exhibit strong performance while being 30x more communication efficient than global distributed training methods like G-AdamW. To broaden our comparison, we introduced two additional baseline methods: **D-SIGNUM (Avg)** and **D-SIGNUM (MaVo)**. These baselines apply our proposed techniques to the SIGNUM framework instead of Lion.[5] We set $\beta = 0.99$ for D-SIGNUM. According to our results, depicted in Figure 4, these SIGNUM-based methods do not perform as well as their Lion-based counterparts.

- We notice that the overall performance of the same optimizer is worse as $k$ is larger, this is consistent with the observation made in DGC [24]. We hypothesize that this may be due to the larger effective batch size resulting in smaller stochasticity, which is consistent with why D-Lion (MaVo) performs a bit better than G-Lion on CIFAR-10 (Figure 3).

---

[5]Note that D-SIGNUM (Avg/MaVo) further subsumes D-SignSGD [5, 6].

## 5.2 Scale to Larger Models on Larger Datasets

To answer **Q3**, we validate Distributed Lion on several large-scale setups including both vision and natural language processing tasks. Under this setting, we compare D-Lion (MaVo) and D-Lion (Avg) against G-AdamW and G-Lion. For the vision task, we tested ViT-S/16 [16] and ViT-B/16 on the ImageNet-1K [31] classification benchmark. For the natural language processing task, we perform both language pretraining and finetuning tasks. This is because Lion has shown good results on language modeling. For the language model pretraining task, we pretrain GPT2++ [29] (the GPT-2 model with modern training techniques adopted from the LLaMA model [35]) on the OpenWebText [17] benchmark, for both 350M and 760M size models. For the language model finetuning task, we conduct few-shot finetuning of the LLaMA 7B model [35] and evaluate the models' downstream performance on standard downstream evaluation benchmarks [13, 37, 12, 27, 7, 32].

**Experiment Setup** For the ImageNet-1K benchmark, we train all methods for 300 epochs, using a global batch size of 4096 and data augmentations MixUp [39] of 0.5 and AutoAug [15]. When training ViT-S/16, we use a learning rate of $3e^{-3}$ for G-AdamW, with betas of $(0.9, 0.999)$ and a weight decay of 0.1. For G-Lion, D-Lion (MaVo), and D-Lion (Avg), we use a learning rate of $3e^{-4}$, betas of $(0.9, 0.99)$, and a weight decay of 1.0. As for ViT-B/16, we use a learning rate of $1e^{-3}$ for G-AdamW, with betas of $(0.9, 0.999)$ and a weight decay of 1.0, while for all Lion variants, we use a learning rate of $1e^{-4}$, betas of $(0.9, 0.99)$, and a weight decay of 10.0. For pretraining language models on the OpenWebText dataset, we build GPT2++ models using the original GPT2 model, but with modern training techniques from the LLaMA model, including using the Gated Linear Unit activation for the multilayer layer perceptron layers (MLPs) and the RMSNorm [38] instead of the LayerNorm [3]. Following the Chinchilla scaling law [18], we trained the 350M model for 14,000 iterations and the 760M model for 30,000 iterations, both with 1,024 tokens. For G-AdamW, we use a learning rate of $3e^{-4}$, betas of $(0.95, 0.99)$, and a weight decay of 0.1. For all Lion variants, we use a learning rate of $9e^{-5}$, betas of $(0.9, 0.99)$, and a weight decay of 1.0. All the models are trained under a global batch size of 480. For the instruction finetuning task, we instruct finetune a LLaMA 7B model for 3 epochs with batch size 32. We use $2e^{-5}$ learning rate, betas of $(0.9, 0.999)$, 0 weight decay for G-AdamW and $6e^{-6}$, $(0.9, 0.99)$ betas, 0.01 weight decay for all Lion variants. For all pretraining experiments, we use $4\text{nodes} \times 8\text{gpus} = 32$ workers. For instruction finetuning experiments, we use 4 workers per experiment.

Table 2: Results on ImageNet classification and OpenWebText language modeling. For ImageNet experiments, we report the Top-1 accuracy. For language modeling experiments, we report the validation perplexity. The best performance is marked with bold text, and the second best with an underline.

| Method | Image Classification | | Language Modeling | |
|---|---|---|---|---|
| | ViT-S/16 | ViT-B/16 | GPT-2++ (350M) | GPT-2++ (760M) |
| AdamW | 79.74 | 80.94 | 18.43 | 14.70 |
| G-Lion | 79.82 | 80.99 | **18.35** | **14.66** |
| D-Lion (MaVo) | 79.69 | 80.79 | 18.37 | **14.66** |
| D-Lion (Avg) | **80.11** | **81.13** | 18.39 | 14.69 |

Table 3: 3-Shot instruction finetuning downstream evaluation results on various datasets. We mark the best performance with bold text and the second one with an underline.

| Method | Arc-Easy | Arc-Challenge | BoolQ | PIQA | SIQA | HellaSwag | OBQA |
|---|---|---|---|---|---|---|---|
| 0-Shot | 76.64 | 43.06 | 76.43 | 78.64 | 45.96 | 56.87 | 33.53 |
| G-AdamW | 77.06 | **46.06** | 77.23 | **79.18** | 48.97 | **59.23** | 35.51 |
| G-Lion | **77.11** | 45.54 | **77.50** | **79.18** | 49.64 | 58.93 | 35.51 |
| D-Lion (MaVo) | 76.86 | 45.72 | 77.14 | 78.92 | **49.75** | 58.96 | **35.71** |
| D-Lion (Avg) | 76.35 | 45.54 | 76.90 | 78.76 | 48.06 | 59.06 | 32.14 |

**Observation** We summarize the results in Table 2 (ImageNet 1K and OpenWebText Language Model Pretraining) and Table 3 (Instruction Finetuning). Both D-Lion (Avg) and D-Lion (MaVo)

can maintain a performance similar to, or even better than, that of G-AdamW and G-Lion, on both large-scale vision and language tasks. We observe that D-Lion (Avg) outperforms D-Lion (MaVo) on ImageNet, and observe the opposite on language modeling and instruction finetuning. We hypothesize that these differences are due to the impact of global batch size. As a result, we recommend using D-Lion (Avg) / (MaVo) when the global batch size is large / small.

## 6   Conclusion and Future Work

In this paper, we introduced Distributed Lion, a communication-efficient distributed training strategy that builds upon the Lion optimizer's binary update mechanism. Distributed Lion is designed to minimize communication overhead by allowing workers to independently manage their optimizer states and exchange only binary or low-precision update vectors with the server. We proposed two aggregation techniques within the Distributed Lion framework: average-based (Distributed Lion Avg) and majority vote-based (Distributed Lion MaVo) algorithms. We provide both theoretical and empirical results to demonstrate Distributed Lion's effectiveness, scalability, and efficiency. Notably, we show that Distributed Lion performs significantly better than existing communication-friendly methods. In the meantime, Distributed Lion demonstrates performance on par with strong global distributed training baselines, while being 32x more communication efficient. As our method is orthogonal to existing communication-efficient methods, an interesting future direction is to combine both techniques for further improvement. As a limitation, currently Distributed Lion (Avg / MaVo) performs inconsistently across different datasets and benchmarks, it will be an interesting future research direction to understand when and why one performs better than the other.

## 7   Acknowledgment

The research is conducted in Statistics & AI group at UT Austin, which receives supports in part from NSF CAREER1846421, SenSE2037267, Office of Navy Research, and NSF AI Institute for Foundations of Machine Learning (IFML).

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

# A  Additional Experiment Details

In this section, we provide additional experiment details.

**CIFAR Experiments**   We list the optimal hyperparameters selected for each method from Figure 2 in Table 4. The learning rates are selected from $\{0.00005, 0.001, 0.005, 0.01\}$ and the weight decays are selected from $\{0.0005, 0.001, 0.005\}$. For each experiment, we use a cosine learning rate scheduler and run for 200 epochs, and we ensure that in each epoch, each local worker sees the entire dataset once.

| Method | LR $\epsilon$ | WD $\lambda$ | Compression Rate |
|---|---|---|---|
| G-AdamW | 0.0001 | 0.0005 | - |
| G-Lion | 0.00005 | 0.005 | - |
| DGC | 0.01 | 0.0005 | 0.96 |
| GradDrop | 0.001 | 0.0005 | 0.96 |
| TernGrad | 0.001 | 0.0005 | - |
| D-Lion (Avg) | 0.00005 | 0.005 | - |
| D-Lion (MaVo) | 0.00005 | 0.005 | - |

Table 4: Hyperparameters for each method in Figure 2. Where LR represents learning rate and WD represents weight decay.

# B  Theory

This section is focusing on the proof of Lion dynamics, and will be organized into these folders:

- Phase I:
    - Constraint enforcing: Discrete time
- Phase II:
    - Majority Voting convergence
    - Avg update convergence
    - Global LION convergence

In line with the behavior observed in the global Lion approach, Lion under a distributed setting also exhibits the two phases. In Section B.1, we show that converging to box can be exponentially fast using our Algorithm 1. We start with introducing a notion of KKT score function that quantifies a stationary solution to the box constrained optimization problem (6) in Section B.2. Building on this, we then proceed to analyze the convergence in terms of the KKT score function for the majority vote (Section B.2.1), averaging (Section B.2.2), and global LION strategies (Section B.2.3).

## B.1  Phase I: Constraint Enforcing

We study phase I in this section. We show that when the constraint is not satisfied, both variants of distributed Lion decrease the distance to the feasible set exponentially fast.

**Theorem B.1** (Phase I). *Assume $f : \mathbb{R}^d \to \mathbb{R}$ is L-smooth, $\beta_1, \beta_2 \in (0, 1)$, and $\beta_2 > \beta_1$, and $\epsilon, \lambda > 0$, and $1 - \epsilon\lambda \in (0, 1)$. Let $(x_t)_{t>0}$ be generated by Algorithm 1. Define $\mathcal{F} = \{x : \|\lambda x\|_\infty \le 1\}$, and $\mathrm{dist}(x_t, \mathcal{F}) = \inf_{z \in \mathcal{F}} \|z - x_t\|$ w.r.t. any norm $\|\cdot\|$.*

*For any two non-negative integers $s \le t$, then $\forall s \le t$, we have*

$$\mathrm{dist}(x_t, \mathcal{F}) \le (1 - \epsilon\lambda)^{t-s}\mathrm{dist}(x_s, \mathcal{F}).$$

*Proof.* Recall Algorithm 1:

$$\delta_{i,t} \leftarrow \text{sign}\big(\beta_1 m_{i,t} + (1 - \beta_1)\nabla_x f(x_t; \xi_{i,t})\big)$$

$$m_{i,t+1} \leftarrow \beta_2 m_{i,t} + (1 - \beta_2)\nabla_x f(x_t; \xi_{i,t})$$

$$\Delta_t = \begin{cases} \frac{1}{N}\big(\sum_{i=1}^N \delta_{i,t}\big) & \text{(Averaging)} \\ \text{sign}\big(\sum_{i=1}^N \delta_{i,t}\big) & \text{(Majority Vote)} \end{cases}$$

$$x_{t+1} = x_t - \epsilon(\Delta_t + \lambda x_t)$$

Rewrite the update into the following form:

$$x_{t+1} = (1 - \epsilon\lambda)x_t - \epsilon\Delta_t,$$

Define $w_{s\to t} = (1 - \epsilon\lambda)^{t-s}$. Unrolling this update yields,

$$x_t = (1 - w_{s\to t})z_{s\to t} + w_{s\to t}x_s, \qquad z_{s\to t} = \frac{\sum_{k=s}^{t-1} w_{k\to t}(-\Delta_t/\lambda)}{\sum_{k=s}^{t-1} w_{k\to t}}.$$

We have $z_{s\to t} \in \mathcal{F}$ since $-\Delta_t/\lambda \in \mathcal{F}$. For any $\epsilon > 0$, let $\hat{x}_s \in \mathcal{F}$ be the point satisfying $\|\hat{x}_s - x_s\| \leq \text{dist}(x_s, \mathcal{F}) + \eta$. Hence, we have

$$\begin{aligned} \text{dist}(x_t, \mathcal{F}) &= \inf_{z \in \mathcal{F}} \|x_t - z\| \\ &\leq \|x_t - (1 - w_{s\to t})z_{s\to t} - w_{s\to t}\hat{x}_s)\| \\ &= w_{s\to t}\|x_s - \hat{x}_s\| \\ &\leq (1 - \epsilon\lambda)^{t-s}(\text{dist}(x_s, \mathcal{F}) + \eta). \end{aligned}$$

As $\eta \to 0$, we achieve the desired result. $\qquad\square$

## B.2  Phase II

We study the convergence of Phase II in this section. We begin by defining a KKT score function to quantify stationary solutions for the box-constrained optimization problem discussed in Section B.2. Following this, we analyze convergence through the KKT score across majority vote (Section B.2.1), averaging (Section B.2.2), and global Lion strategies (Section B.2.3).

First, we list the following assumptions used in our proof.

**Assumption B.2** (Smooth and Differentiable $f$). *Function $f(\cdot)$ is differentiable and $L$-smooth.*

**Assumption B.3** (Variance bound). *$\mathcal{D}_i$ is i.i.d. drawn from a common distribtion $\pi_*$, and the stochastic sample $\xi^i \sim \mathcal{D}_i$ is i.i.d. and upon receiving query $x \in \mathbb{R}^d$, the stochastic gradient oracle gives us an* independent *unbiased estimate $\nabla f(x; \xi^i)$ from the $i$-th worker that has coordinate bounded variance:*

$$\mathbb{E}_\xi[\nabla f(x; \xi^i)] = \nabla f(x), \qquad \mathbb{E}_\xi\left[\|\nabla f(x; \xi^i) - \nabla f(x)\|^2\right] \leq \sigma^2.$$

**Assumption B.4** (Bias Correction). *Consider the sequence $\{m_t^i\}_{t>0, i\in[N]}$ generated by Algorithm 1, $\mathbb{E}[\tilde{m}_t^i]/\mathbb{E}[\text{sign}(\tilde{m}_t^i)] \geq 0$.*

Here we define the a KKT score function for box constrained problem (6):

$$\mathcal{S}(x) := \langle \nabla f(x), \text{sign}(\nabla f(x)) + \lambda x \rangle.$$

**Proposition B.5.** *Assume $f$ is continuously differentiable, $\lambda > 0$, and $\|\lambda x\|_\infty \leq 1$. Then $\mathcal{S}(x) = 0$ implies a KKT stationary condition of $\min_x f(x)$ s.t. $\|\lambda x\|_\infty \leq 1$.*

*Proof.* We will verify that $\mathcal{S}(x) = 0$ coincides with the first order KKT conditions of the box constrained optimization problem (6).

Recall the box constrained problem in (6), we can rewrite it into the following formulation:

$$\min_{x \in \mathbb{R}^d} f(x) \quad s.t. \quad \lambda x_i - 1 \leq 0, \quad -\lambda x_i - 1 \leq 0, \quad \forall\, i \in [d].$$

Let $\mu = (\mu_1, \mu_2, \cdots, \mu_d)^\top$ and $\tilde{\mu} = (\tilde{\mu}_1, \tilde{\mu}_2, \cdots, \tilde{\mu}_d)^\top$, then its first order KKT stationary condition can be written as:

$$\partial_{x_i} f(x) + \mu_i \lambda - \tilde{\mu}_i \lambda = 0 \qquad\qquad \text{//Stationarity}$$
$$\mu_i(\lambda x_i - 1) = 0, \quad \tilde{\mu}_i(-\lambda x_i - 1) = 0 \qquad \text{//Complementary slackness}$$
$$\mu_i \geq 0, \quad \tilde{\mu}_i \geq 0 \qquad\qquad\qquad \text{//Dual feasibility}$$
$$\lambda x_i - 1 \leq 0, \quad -\lambda x_i - 1 \leq 0 \qquad\qquad \text{//Primal feasibility}$$
$$\forall\, i \in \{1, 2, \cdots, d\}.$$

Expressing $\mathcal{S}(x)$ element-wisely, we obtain:

$$\mathcal{S}(x) = \sum_{k=1}^d \mathcal{S}_k(x), \qquad \text{with} \qquad \mathcal{S}_k(x) = \partial_{x_k} f(x) \cdot (\text{sign}(\partial_{x_k} f(x)) + \lambda x_k),$$

where $x_k$ denotes the $k$-th element of vector $x$. Since $\|\lambda x\|_\infty \leq 1$, we have $\mathcal{S}_k(x) \geq 0$, because

$$\begin{aligned}
\mathcal{S}_k(x) &= \partial_{x_k} f(x) \cdot (\text{sign}(\partial_{x_k} f(x)) + \lambda x_k) \\
&= |\partial_{x_k} f(x)| + \lambda \partial_{x_k} f(x) \cdot x_k \\
&\geq |\partial_{x_k} f(x)| - |\partial_{x_k} f(x)| \cdot |\lambda x_k| \\
&= |\partial_{x_k} f(x)|(1 - |\lambda x_k|) \\
&\geq 0 \qquad \text{//since } \|\lambda x\|_\infty \leq 1.
\end{aligned}$$

Hence, if $\mathcal{S}(x) = 0$, we have $\mathcal{S}_k(x) = 0$ for each component $k$. It means that we have either $\text{sign}(\partial_{x_k} f(x)) + \lambda x_k = 0$ or $\partial_{x_k} f(x) = 0$ for each coordinate $k$.

There are two primary cases to consider for each $k$:

- **Case I**: $\partial_{x_k} f(x) = 0$. This suggests that we reach a stationary condition of $f(x)$ w.r.t. coordinate $x_k$, and the KKT condition is satisfied in this case with $\mu_k = \tilde{\mu}_k = 0$.

- **Case II**: $\text{sign}(\partial_{x_k} f(x)) + \lambda x_k = 0$, it follows that $x_k = -\frac{1}{\lambda}\text{sign}(\partial_{x_k} f(x))$.
  - if $\text{sign}(\partial_{x_k} f(x) = 1$, then $\partial_{x_k} f(x) \geq 0$, and the KKT condition is satisfied with $\mu_k = 0$ and $\tilde{\mu}_k = \partial_{x_k} f(x)/\lambda$
  - if $\text{sign}(\partial_{x_k} f(x)) = -1$, then $\partial_{x_k} f(x) \leq 0$, and the KKT condition is satisfied with $\tilde{\mu}_k = 0$ and $\mu_k = \partial_{x_k} f(x)/\lambda$.

It turns out the two cases above exactly covers the KKT stationary solution pair $(x, \mu, \tilde{\mu})$ of the box constrained problem in (6).

In conclusion, $\mathcal{S}(x) = 0$ signifies reaching a stationary point of the bound-constrained optimization problem, as formulated in (6), providing critical insights into the convergence behavior of the algorithm under consideration. $\qquad\square$

### B.2.1 Majority Vote

Assume $f \colon \mathbb{R}^d \to \mathbb{R}$ is $L$-smooth, and $N$ is the number of workers, on the $i$-th worker, consider the following scheme based on the majority vote:

$$\begin{aligned}
g_t^i &:= \nabla f(x_t; \xi_t^i) \\
m_{t+1}^i &= \beta_2 m_t^i + (1 - \beta_2)g_t^i \\
\tilde{m}_{t+1}^i &= \beta_1 m_t^i + (1 - \beta_1)g_t^i \\
x_{t+1} &= x_t - \epsilon \left( \text{sign}\left( \sum_{i=1}^N \text{sign}(\tilde{m}_{t+1}^i) \right) + \lambda x_t \right). \qquad \text{//Majority Voting}
\end{aligned} \tag{11}$$

**Theorem B.6** (Convergence in Phase II). *Assumption B.2 B.3 B.4 hold, consider the scheme in Algorithm 11, and $\beta_1, \beta_2 \in (0,1)$, and $\beta_2 > \beta_1$, and $\epsilon, \lambda > 0$. $\|\lambda x_0\|_\infty \leq 1$.*

*We have*

$$\frac{1}{T}\sum_{t=1}^{T}\mathbb{E}\mathcal{S}(x_t) \leq \frac{f(x_0) - f^*}{T\epsilon} + \frac{2D\beta_1\beta_2\sqrt{d}\|\nabla f(x_0)\|}{T(1-\beta_2)} + \frac{4\beta_1 L\epsilon d}{1-\beta_2} + \frac{2\sqrt{d}\sigma(1+\sqrt{C}) + 2\rho}{\sqrt{N}} + 2L\epsilon d,$$

*where $C = \beta_1^2(1-\beta_2)\frac{1}{1+\beta_2} + (1-\beta_1)^2$, $D = \max\{1, \sigma / \left(2\sqrt{d}\beta_1\beta_2^T\|\nabla f(x_0)\|\right)\}$, and*

$$\rho_t[k] = \begin{cases} 0 & \text{if } \mathbb{E}[\text{sign}(\tilde{m}_{t+1}^i[k])] = 0, \\ \mathbb{E}[\tilde{m}_{t+1}^i[k]]/\mathbb{E}[\text{sign}(\tilde{m}_{t+1}^i[k])] & \text{else.} \end{cases}$$

*Proof.* Following Theorem B.1 from phase 1, once we have $\|\lambda x_0\|_\infty \leq 1$, we stay within the constraint set with $\|\lambda x_t\| \leq 1$ for all subsequent time $t \geq 0$.

For notation, write $\tilde{M}_{t+1} = \sum_{i=1}^{N}\text{sign}(\tilde{m}_{t+1}^i)$. This yields $x_{t+1} = x_t - \epsilon\text{sign}(\tilde{M}_{t+1}) - \epsilon\lambda x_t$. We have

$$
\begin{aligned}
f(x_{t+1}) - f(x_t) &\leq \langle \nabla f(x_t), x_{t+1} - x_t \rangle + \frac{L}{2}\|x_{t+1} - x_t\|_2^2 \qquad \textcolor{magenta}{//L\text{-smoothness of } f} \\
&= -\epsilon\langle \nabla f(x_t), \text{sign}(\tilde{M}_{t+1}) + \lambda x_t \rangle + \frac{L}{2}\|x_{t+1} - x_t\|_2^2 \\
&= -\epsilon\langle \nabla f(x_t), \text{sign}(\nabla f(x_t)) + \lambda x_t \rangle + \frac{L}{2}\|x_{t+1} - x_t\|_2^2 \\
&\quad + \epsilon\langle \nabla f(x_t), \text{sign}(\nabla f(x_t)) - \text{sign}(\tilde{M}_{t+1}) \rangle \\
&\leq -\epsilon\mathcal{S}(x_t) + 2L\epsilon^2 d + \epsilon\langle \nabla f(x_t), \text{sign}(\nabla f(x_t)) - \text{sign}(\tilde{M}_{t+1}) \rangle, \qquad (12)
\end{aligned}
$$

where we used $\|x_{t+1} - x_t\|^2 = \epsilon^2 \left\|\text{sign}(\tilde{M}_{t+1}) + \lambda x_t\right\|^2 \leq 4\epsilon^2 d$, because $\|\lambda x_t\|_\infty \leq 1$.

By Assumption B.3, $\tilde{m}_{t+1}^1, \tilde{m}_{t+1}^2, \cdots, \tilde{m}_{t+1}^N$ are i.i.d., so $\mathbb{E}[\tilde{m}_{t+1}^i]$ and $\mathbb{E}[\text{sign}(\tilde{m}_{t+1}^i)]$ don't depend on $i$. Hence we can define $R_{t+1} = \mathbb{E}[\tilde{m}_{t+1}^i]/\mathbb{E}[\text{sign}(\tilde{m}_{t+1}^i)]$, where the division operation is element wise, so $R_{t+1} \in \mathbb{R}^d$.

By Assumption 3.3, $R_t$ is non-negative, one special case for the ratio $R_t$ is when $\mathbb{E}[\text{sign}(\tilde{m}_t^i[k])] = 0$, yet $\mathbb{E}[\tilde{m}_t^i[k]] \neq 0$, leading to $R_t[k] = +\infty$ for $k \in [d]$. In such instance, $P(\tilde{m}_t^i[k] > 0) = 1/2$ derived from the equation $\mathbb{E}[\text{sign}(\tilde{m}_t^i[k])] = 2P(\tilde{m}_t^i[k] > 0) - 1 = 0$, for $k \in [d]$.

First, recognizing that $\mathbb{E}[\text{sign}(\tilde{M}_t[k])] = 0$ is straightforward as we model it as a binomial distribution with success probability $p = 1/2$ for $t > 0$. This leads to the result $\mathbb{E}\nabla f(x_t)[k]\left(\text{sign}(\nabla f(x_t)[k]) - \text{sign}(\tilde{M}_t[k])\right) = \mathbb{E}|\nabla f(x_t)[k]|$.

Given that $\mathbb{E}[X] = \arg\min_z \mathbb{E}\|X - z\|_2$ defines the expectation of a random variable $X$ as the value $z$ minimizes the expected euclidean distance to $X$, and the *median* $X = \arg\min_z \mathbb{E}\|X - z\|_1$ defines the median as the value $z$ minimizing the expected absolute distance to $X$, for a R.V. $X$ in $\mathbb{R}$, recall our case where $P(\tilde{m}_t^i[k] > 0) = 1/2$, which is equivalent to that the median is 0. From this, it follows that

$$\mathbb{E}|\nabla f(x_t)[k]| \leq \mathbb{E}[\mathbb{E}_\xi[\left|\nabla f(x_t; \xi_t^i)[k] - \nabla f(x_t)[k]\right|_1]] \leq \mathbb{E}\sqrt{\mathbb{E}_\xi\left\|\nabla f(x_t; \xi_t^i)[k] - \nabla f(x_t)[k]\right\|_2^2} \leq \sigma.$$

To bound the last term in (12) $\langle \nabla f(x_t), \text{sign}(\nabla f(x_t)) - \text{sign}(\tilde{M}_{t+1}) \rangle$, we follow a structured approach. Here's an outline for bounding this term:

To bound the last term in Equation (12), $\langle \nabla f(x_t), \text{sign}(\nabla f(x_t)) - \text{sign}(\tilde{M}_{t+1}) \rangle$, we follow a structured approach:

1. **Transform Inner Product into Norm of Difference**: Using Lemma B.8 to convert the inner product $\langle \nabla f(x_t), \text{sign}(\nabla f(x_t)) - \text{sign}(\tilde{M}_{t+1}) \rangle$ into the norm of a difference.

2. **Introduce $R_t$ as a De-bias Ratio**: $R_t$ is defined to adjust or correct for any bias in the expected value of $\tilde{m}_t^i$ and the expected sign of $\tilde{m}_t^i$ as in Assumption B.4.

3. **Handle Cases of $R_t$ Separately**: Given the possibility of $R_t[k] = +\infty$, it's essential to treat the scenarios of $R_t[k] < +\infty$ and $R_t[k] = +\infty$ with separate proofs.

   - For $R_t[k] < +\infty$, standard bounding techniques can be applied, potentially leveraging properties of $R_t$ to establish a finite upper bound.

   - For $R_t[k] = +\infty$, it's actually bounding $\|\nabla f(x_t)\|$. This can be bounded by the variance of the stochastic gradient $g_t^i$.

4. **Merge Cases with Finite $\rho_t$ Replacing $R_t$**: After separately proving bounds for each case of $R_t$, the results are unified by substituting $R_t$ with a finite $\rho_t$, where $\rho_t$ serves a similar purpose but ensures a manageable, finite adjustment.

**Case I (Finite $R_{t+1}$)**

The first step is to expand this inner product, we have

$$
\mathbb{E}\langle \nabla f(x_t), \text{sign}(\nabla f(x_t)) - \text{sign}(\tilde{M}_{t+1})\rangle
$$

$$
= \mathbb{E}\langle \nabla f(x_t), \text{sign}(\nabla f(x_t)) - \text{sign}(\frac{1}{N}\tilde{M}_{t+1})\rangle
$$

$$
= \mathbb{E}\sum_{k=1}^{d} \nabla f(x_t)[k]\left(\text{sign}(\nabla f(x_t)[k]) - \text{sign}(\frac{1}{N}\tilde{M}_{t+1}[k])\right)
$$

$$
= 2\mathbb{E}\sum_{k=1}^{d} R_{t+1}[k]\left|\nabla f(x_t)[k]/R_{t+1}[k] - \frac{1}{N}\tilde{M}_{t+1}[k]\right|
$$

$$
= 2\mathbb{E}\sum_{k=1}^{d} R_{t+1}[k]\left|\nabla f(x_t)[k]/R_{t+1}[k] - \frac{1}{N}\sum_{i=1}^{N}\text{sign}(\tilde{m}_{t+1}^i[k])\right|. \qquad \text{//Lemma B.8 and Assumption 3.3}
$$

By definition of $R_t$, it is a debiasing ratio between $\mathbb{E}[\tilde{m}_{t+1}^i]$ and $\mathbb{E}[\text{sign}(\tilde{m}_{t+1}^i)]$, so we construct a difference between $\frac{1}{N}\sum_{i=1}^{N}\text{sign}(\tilde{m}_{t+1}^i[k])$ and $\frac{1}{N}\sum_{i=1}^{N}\tilde{m}_{t+1}^i[k]$ by decoupling the difference between $\nabla f(x_t)[k]/R_{t+1}[k]$ and $\frac{1}{N}\text{sign}(\tilde{m}_{t+1}^i[k])$.

$$
\mathbb{E}R_{t+1}[k]\left|\nabla f(x_t)[k]/R_{t+1}[k] - \frac{1}{N}\sum_{i=1}^{N}\text{sign}(\tilde{m}_{t+1}^i[k])\right|
$$

$$
= \mathbb{E}R_{t+1}[k]\left|\nabla f(x_t)[k]/R_{t+1}[k] - \frac{1}{N}\sum_{i=1}^{N}\tilde{m}_{t+1}^i[k]/R_{t+1}[k] + \frac{1}{N}\sum_{i=1}^{N}\tilde{m}_{t+1}^i[k]//R_{t+1}[k] - \frac{1}{N}\sum_{i=1}^{N}\text{sign}(\tilde{m}_{t+1}^i[k])\right|
$$

$$
= \mathbb{E}R_{t+1}[k]\left|\nabla f(x_t)[k]/R_{t+1}[k] - \frac{1}{N}\sum_{i=1}^{N}\tilde{m}_{t+1}^i[k]/R_{t+1}[k]\right| + R_{t+1}[k]\left|\frac{1}{N}\sum_{i=1}^{N}\tilde{m}_{t+1}^i[k]/R_{t+1}[k] - \frac{1}{N}\sum_{i=1}^{N}\text{sign}(\tilde{m}_{t+1}^i[k])\right|
$$

$$
= \mathbb{E}\left|\nabla f(x_t)[k] - \frac{1}{N}\sum_{i=1}^{N}\tilde{m}_{t+1}^i[k]\right| + R_{t+1}[k]\left|\frac{1}{N}\sum_{i=1}^{N}\tilde{m}_{t+1}^i[k]/R_{t+1}[k] - \frac{1}{N}\sum_{i=1}^{N}\text{sign}(\tilde{m}_{t+1}^i[k])\right|.
$$

The first term $\mathbb{E}\left|\nabla f(x_t)[k] - \frac{1}{N}\sum_{i=1}^{N}\tilde{m}_{t+1}^i[k]\right|$ doesn't depend on $R_{t+1}$, we can bound this term across $d$ coordinates using Lemma B.10:

$$\mathbb{E}\sum_{k=1}^{d}\left|\nabla f(x_t)[k] - \frac{1}{N}\sum_{i=1}^{N}\tilde{m}_{t+1}^i[k]\right| \leq \sqrt{d}\,\mathbb{E}\left\|\nabla f(x_t) - \frac{1}{N}\sum_{i=1}^{N}\tilde{m}_{t+1}^i\right\|$$

$$\leq \sqrt{d}\,\mathbb{E}\left\|\nabla f(x_t) - \frac{1}{N}\sum_{i=1}^{N}\left(\beta_1 m_t^i + (1-\beta_1)g_t^i\right)\right\|$$

$$\leq \sqrt{d}\,\mathbb{E}\left\|\frac{1}{N}\sum_{i=1}^{N}\beta_1\left(\nabla f(x_t) - m_t^i\right)\right\| + \left\|\frac{1}{N}\sum_{i=1}^{N}(1-\beta_1)\left(\nabla f(x_t) - g_t^i\right)\right\|$$

$$\leq \sqrt{d}\beta_1\left(\beta_2^t\|\nabla f(x_0)\| + \frac{2L\epsilon\sqrt{d}}{1-\beta_2} + \frac{\sigma}{\sqrt{N(1+\beta_2)}}\right) + \frac{\sqrt{d}\sigma(1-\beta_1)}{\sqrt{N}}. \qquad \text{//Lemma B.}$$

The second term $\mathbb{E}R_{t+1}[k]\left|\frac{1}{N}\sum_{i=1}^{N}\tilde{m}_{t+1}^i[k]/R_{t+1}[k] - \frac{1}{N}\sum_{i=1}^{N}\mathrm{sign}(\tilde{m}_{t+1}^i[k])\right|$ can be decoupled into the variance of $\frac{1}{N}\sum_{i=1}^{N}\mathrm{sign}(\tilde{m}_{t+1}^i[k])$ and the variance of $\frac{1}{N}\sum_{i=1}^{N}\tilde{m}_{t+1}^i[k]$:

$$\mathbb{E}\sum_{k=1}^{d}R_{t+1}[k]\left|\frac{1}{N}\sum_{i=1}^{N}\tilde{m}_{t+1}^i[k]/R_{t+1}[k] - \frac{1}{N}\sum_{i=1}^{N}\mathrm{sign}(\tilde{m}_{t+1}^i[k])\right|$$

$$= \mathbb{E}\sum_{k=1}^{d}R_{t+1}[k]\left|\frac{1}{N}\sum_{i=1}^{N}\tilde{m}_{t+1}^i[k]/R_{t+1}[k] - \mathbb{E}\tilde{m}_{t+1}^i[k]/R_{t+1}[k] + \mathbb{E}\tilde{m}_{t+1}^i[k]/R_{t+1}[k] - \frac{1}{N}\sum_{i=1}^{N}\mathrm{sign}(\tilde{m}_{t+1}^i[k])\right|$$

$$= \mathbb{E}\sum_{k=1}^{d}R_{t+1}[k]\left|\frac{1}{N}\sum_{i=1}^{N}\tilde{m}_{t+1}^i[k]/R_{t+1}[k] - \mathbb{E}\tilde{m}_{t+1}^i[k]/R_{t+1}[k] + \mathbb{E}\mathrm{sign}(\tilde{m}_{t+1}^i[k]) - \frac{1}{N}\sum_{i=1}^{N}\mathrm{sign}(\tilde{m}_{t+1}^i[k])\right|$$

$$= \mathbb{E}\sum_{k=1}^{d}R_{t+1}[k]\left|\frac{1}{N}\sum_{i=1}^{N}\tilde{m}_{t+1}^i[k]/R_{t+1}[k] - \mathbb{E}\tilde{m}_{t+1}^i[k]/R_{t+1}[k]\right| + R_{t+1}[k]\left|\mathbb{E}\mathrm{sign}(\tilde{m}_{t+1}^i[k]) - \frac{1}{N}\sum_{i=1}^{N}\mathrm{sign}(\tilde{m}_{t+1}^i[k])\right|$$

$$= \mathbb{E}\sum_{k=1}^{d}\left|\frac{1}{N}\sum_{i=1}^{N}\tilde{m}_{t+1}^i[k] - \mathbb{E}\tilde{m}_{t+1}^i[k]\right| + R_{t+1}[k]\left|\frac{1}{N}\sum_{i=1}^{N}\mathrm{sign}(\tilde{m}_{t+1}^i) - \mathbb{E}\mathrm{sign}(\tilde{m}_{t+1}^i)\right|$$

$$\leq \mathbb{E}\sqrt{d}\left\|\frac{1}{N}\sum_{i=1}^{N}\tilde{m}_{t+1}^i - \mathbb{E}\tilde{m}_{t+1}^i\right\| + \|R_{t+1}\|\left\|\frac{1}{N}\sum_{i=1}^{N}\mathrm{sign}(\tilde{m}_{t+1}^i) - \mathbb{E}\mathrm{sign}(\tilde{m}_{t+1}^i)\right\|.$$

Now we have got the variance of $\frac{1}{N}\sum_{i=1}^{N}\mathrm{sign}(\tilde{m}_{t+1}^i[k])$ and the variance of $\frac{1}{N}\sum_{i=1}^{N}\tilde{m}_{t+1}^i[k]$, let us bound them one by one:

**The variance of $\frac{1}{N}\sum_{i=1}^{N}\tilde{m}_{t+1}^i[k]$**

$$\sqrt{d}\,\mathbb{E}\left\|\frac{1}{N}\sum_{i=1}^{N}\tilde{m}_{t+1}^i - \mathbb{E}\tilde{m}_{t+1}^i\right\| \leq \sqrt{d}\sqrt{\mathbb{E}\left\|\frac{1}{N}\sum_{i=1}^{N}\tilde{m}_{t+1}^i - \mathbb{E}\tilde{m}_{t+1}^i\right\|^2}$$

$$= \sqrt{d}\sqrt{\frac{1}{N^2}\sum_{i=1}^{N}\mathbb{E}\left\|\tilde{m}_{t+1}^i - \mathbb{E}\tilde{m}_{t+1}^i\right\|^2}$$

$$\leq \sqrt{\frac{Cd\sigma^2}{N}}, \qquad \text{//Lemma B.11}$$

where $C = \beta_1^2(1-\beta_2)\frac{1}{1+\beta_2} + (1-\beta_1)^2$.

**The variance of** $\frac{1}{N} \sum_{i=1}^N \operatorname{sign}(\tilde{m}_{t+1}^i[k])$

$$\|R_{t+1}\| \, \mathbb{E} \left\| \frac{1}{N} \sum_{i=1}^N \operatorname{sign}(\tilde{m}_{t+1}^i) - \mathbb{E}\operatorname{sign}(\tilde{m}_{t+1}^i) \right\| \leq \sqrt{\mathbb{E} \left\| \sum_{i=1}^N \operatorname{sign}(\tilde{m}_{t+1}^i)/N - \mathbb{E}[\operatorname{sign}(\tilde{m}_{t+1}^i)] \right\|^2}$$

$$= \|R_{t+1}\| \sqrt{\frac{1}{N^2} \sum_{i=1}^N \mathbb{E} \left\| \operatorname{sign}(\tilde{m}_{t+1}^i) - \mathbb{E}[\operatorname{sign}(\tilde{m}_{t+1}^i)] \right\|^2}$$

$$\leq \|R_{t+1}\| \sqrt{\frac{1}{N}}. \qquad \textcolor{magenta}{//\text{Lemma B.9}}$$

In above, we have the bound of the last term in (12) $\langle \nabla f(x_t), \operatorname{sign}(\nabla f(x_t)) - \operatorname{sign}(\tilde{M}_{t+1}) \rangle$:

$$\mathbb{E}\langle \nabla f(x_t), \operatorname{sign}(\nabla f(x_t)) - \operatorname{sign}(\tilde{M}_{t+1}) \rangle$$

$$\leq 2\mathbb{E} \sum_{k=1}^d \left| \nabla f(x_t)[k] - \frac{1}{N} \sum_{i=1}^N \tilde{m}_{t+1}^i[k] \right| + 2\mathbb{E} \sum_{k=1}^d R_{t+1}[k] \left| \frac{1}{N} \sum_{i=1}^N \tilde{m}_{t+1}^i[k]/R_{t+1}[k] - \frac{1}{N} \sum_{i=1}^N \operatorname{sign}(\tilde{m}_{t+1}^i[k]) \right|$$

$$\leq 2\sqrt{d}\mathbb{E} \left\| \nabla f(x_t) - \frac{1}{N} \sum_{i=1}^N \tilde{m}_{t+1}^i \right\| + 2\mathbb{E}\sqrt{d} \left\| \frac{1}{N} \sum_{i=1}^N \tilde{m}_{t+1}^i - \mathbb{E}\tilde{m}_{t+1}^i \right\| + 2\|R_{t+1}\| \left\| \frac{1}{N} \sum_{i=1}^N \operatorname{sign}(\tilde{m}_{t+1}^i) - \mathbb{E}\operatorname{sign}(\tilde{m}_{t+1}^i) \right\|$$

$$\leq 2\sqrt{d}\beta_1 \left( \beta_2^t \|\nabla f(x_0)\| + \frac{2L\epsilon\sqrt{d}}{1 - \beta_2} + \frac{\sigma}{\sqrt{N(1+\beta_2)}} \right) + 2\frac{\sqrt{d}\sigma(1-\beta_1)}{\sqrt{N}} + 2\sqrt{\frac{Cd\sigma^2}{N}} + 2\|R_{t+1}\| \sqrt{\frac{1}{N}}.$$

**Case II (Infinite $R$)**

From our discussion above, we know that $P(\tilde{m}_t^i[k] > 0) = 1/2$ since $\mathbb{E}[\operatorname{sign}(\tilde{m}_t^i[k])] = 2P(\tilde{m}_t^i[k] > 0) - 1 = 0$, where $k \in [d]$. For notion, write $\mathcal{D} = \{j \in [d] \mid \mathbb{E}[\operatorname{sign}(\tilde{m}_{t+1}^i[j])] = 0\}$. In this case, we have

$$\mathbb{E} \sum_{j \in \mathcal{D}} \nabla f(x_t)[j] \left( \operatorname{sign}(\nabla f(x_t)[j]) - \operatorname{sign}(\tilde{M}_t[j]) \right) = \mathbb{E} \sum_{j \in \mathcal{D}} |\nabla f(x_t)[j]|$$

$$\leq \mathbb{E} \left[ \mathbb{E}_\xi \sum_{j \in \mathcal{D}} |\nabla f(x_t; \xi_t^i)[j] - \nabla f(x_t)[j]| \right]$$

$$\leq \mathbb{E} \sqrt{\mathbb{E}_\xi \sum_{j \in \mathcal{D}} \left\| \nabla f(x_t; \xi_t^i)[j] - \nabla f(x_t)[j] \right\|_2^2}$$

$$\leq \sigma.$$

So, the inner product $\langle \nabla f(x_t), \operatorname{sign}(\nabla f(x_t)) - \operatorname{sign}(\tilde{M}_{t+1}) \rangle$ is still bounded. Hence we can merge both cases into a unified bound by simply replacing $R_t$ by $\rho_t$:

$$\rho_t[k] = \begin{cases} 0 & \text{if } \mathbb{E}[\operatorname{sign}(\tilde{m}_{t+1}^i[k])] = 0, \\ \mathbb{E}[\tilde{m}_{t+1}^i[k]]/\mathbb{E}[\operatorname{sign}(\tilde{m}_{t+1}^i[k])] & \text{else.} \end{cases}$$

Adding one constant $D \geq 1$ to make the bound in finite case adpative to infinite case:

$$\sigma \leq 2D\sqrt{d}\beta_1\beta_2^t \|\nabla f(x_0)\|, \forall t, 1 \leq t \leq T.$$

Hence,

$$\mathbb{E} \sum_{j \in \mathcal{D}} \nabla f(x_t)[j] \left( \operatorname{sign}(\nabla f(x_t)[j]) - \operatorname{sign}(\tilde{M}_t[j]) \right)$$

$$\leq 2D\sqrt{d}\beta_1\beta_2^t \|\nabla f(x_0)\| + \frac{4Ld\beta_1\epsilon}{1 - \beta_2} + \frac{2\sqrt{d}\sigma(1 + \sqrt{C}) + 2\|\rho_{t+1}\|}{\sqrt{N}}.$$

Finally, we have the bound for both cases:

$$\mathbb{E}\langle \nabla f(x_t), \text{sign}(\nabla f(x_t)) - \text{sign}(\tilde{M}_{t+1})\rangle$$

$$\leq 2\sqrt{d}\beta_1 \left( \beta_2^t \|\nabla f(x_0)\| + \frac{2L\epsilon\sqrt{d}}{1-\beta_2} + \frac{\sigma}{\sqrt{N(1+\beta_2)}} \right) + 2\frac{\sqrt{d}\sigma(1-\beta_1)}{\sqrt{N}} + 2\sqrt{\frac{Cd\sigma^2}{N}} + 2\|\rho_{t+1}\| \sqrt{\frac{1}{N}}$$

$$\leq 2D\sqrt{d}\beta_1\beta_2^t\|\nabla f(x_0)\| + \frac{4Ld\beta_1\epsilon}{1-\beta_2} + \frac{2\sqrt{d}\sigma(1+\sqrt{C})+2\|\rho_{t+1}\|}{\sqrt{N}}.$$

Then we have

$$f(x_{t+1}) - f(x_t) \leq -\epsilon\mathcal{S}(x_t) + 2L\epsilon^2 d + \epsilon\langle\nabla f(x_t), \text{sign}(\nabla f(x_t)) - \text{sign}(\tilde{M}_{t+1})\rangle$$

$$\leq -\epsilon\mathcal{S}(x_t) + 2L\epsilon^2 d + \epsilon\left( 2D\sqrt{d}\beta_1\beta_2^t\|\nabla f(x_0)\| + \frac{4Ld\beta_1\epsilon}{1-\beta_2} + \frac{2\sqrt{d}\sigma(1+\sqrt{C})+2\|\rho_{t+1}\|}{\sqrt{N}} \right),$$

Hence, a telescope yields

$$\frac{1}{T}\sum_{t=1}^{T}\mathbb{E}\mathcal{S}(x_t) \leq \frac{f(x_0)-f^*}{T\epsilon} + \frac{2D\beta_1\beta_2\sqrt{d}\|\nabla f(x_0)\|}{T(1-\beta_2)} + \frac{4\beta_1 Led}{1-\beta_2} + \frac{2\sqrt{d}\sigma(1+\sqrt{C})+2\rho}{\sqrt{N}} + 2L\epsilon d,$$

where $\rho = \max_{1\leq t\leq T}\|\rho_t\|$. $\qquad\square$

**Lemma B.7.** *Let $(X, Y)$ is a joint random variable on $\mathbb{R}^d \times \mathbb{R}^d$. For any constant $a \in (0, +\infty)$, we have*

$$\mathbb{E}[\langle X, \text{sign}(X) - \text{sign}(Y)\rangle] \leq 2a\sqrt{d}\mathbb{E}\|X/a - Y\|.$$

*Proof.* Without loss of generality, set $a = 1$.

$$\mathbb{E}[\langle X, \text{sign}(X) - \text{sign}(Y)\rangle] = \mathbb{E}[\|X\|_1 - \langle X, \text{sign}(Y)\rangle]$$
$$\leq 2\mathbb{E}[\|X - Y\|_1] \qquad \text{//Lemma B.8}$$
$$\leq 2\sqrt{d}\mathbb{E}[\|X - Y\|] \qquad \text{//by Cauchy-Schwarz,}$$

where $\|\cdot\|_1$ is the $\ell_1$ norm and $\|\cdot\|$ denotes the Euclidean norm. $\qquad\square$

**Lemma B.8.** *For any $x, y \in \mathbb{R}$, we have*

$$|x| - x\text{sign}(y) \leq 2|x - y|.$$

*Proof.* If $\text{sign}(y) = \text{sign}(x)$, we have $|x| - x\text{sign}(y) = 0 \leq 2|x - y|$.
If $\text{sign}(y) = -\text{sign}(x)$, we have $|x| - x\text{sign}(y) = 2|x| \leq 2|x| + 2|y| = 2|x - y|$.
If $\text{sign}(y) = 0$, we have $|x| - x\text{sign}(y) = |x| = |x - y| \leq 2|x - y|$. $\qquad\square$

**Lemma B.9.** *Let $X$ be a random variable in $\mathbb{R}$, we have $\mathbb{E}\|\text{sign}(X) - \mathbb{E}[\text{sign}(X)]\|^2 < 1$.*

*Proof.* The result is a direct derivation from Bernoulli distribution's variance,

$$\mathbb{E}\|\text{sign}(X) - \mathbb{E}[\text{sign}(X)]\|^2 = \mathbb{E}[\text{sign}(X)^2] - \mathbb{E}[\text{sign}(X)]^2 < 1.$$

$\qquad\square$

**Lemma B.10.** *Following the same setting in Theorem B.6, we have*

$$\|\frac{1}{N}\sum_{i=1}^{N}m_t^i - \nabla f(x_t)\| \leq \beta_2^t\|\nabla f(x_0)\| + \frac{2L\varepsilon\sqrt{d}}{1-\beta_2} + \frac{\sigma}{\sqrt{N(1+\beta_2)}}.$$

*Proof.* We use the notions: $g_t^i := \nabla f(x_t; \xi_t^i)$, $M_t = \frac{1}{N}\sum_{i=1}^{N} m_t^i$, $\varepsilon_t := M_t - \nabla f(x_t)$, $\overline{g}_t = \frac{1}{N}\sum_{i=1}^{N} g_t^i$, $\delta_t := \overline{g}_t - \nabla f(x_t)$, and $s_t = \nabla f(x_{t-1}) - \nabla f(x_t)$

$$\begin{aligned}
\varepsilon_t &= M_t - \nabla f(x_t) \\
&= \beta_2 M_{t-1} + (1-\beta_2)\overline{g}_t - \nabla f(x_t) \\
&= \beta_2(M_{t-1} - \nabla f(x_{t-1})) + (1-\beta_2)(\overline{g}_t - \nabla f(x_t)) + \beta_2(\nabla f(x_{t-1}) - \nabla f(x_t)) \\
&= \beta_2 \varepsilon_{t-1} + (1-\beta_2)\delta_t + \beta_2 s_t.
\end{aligned}$$

That is

$$\varepsilon_t = \beta_2 \varepsilon_{t-1} + (1-\beta_2)\delta_t + \beta_2 s_t.$$

Under the $L$-smoothness assumption B.2:

$$\|s_t\| = \|\nabla f(x_{t-1}) - \nabla f(x_t)\| \le L\|x_{t-1} - x_t\| \le 2L\sqrt{d}\epsilon, \tag{13}$$

where $\varepsilon$ is the step size. Using mathematical induction, we have

$$\varepsilon_t = \beta_2^t \varepsilon_0 + \sum_{i=1}^{t} \beta_2^{t-i+1} s_i + (1-\beta_2)\sum_{i=1}^{t} \beta_2^{t-i}\delta_t. \tag{14}$$

By taking the norms of both sides of the above equation and using the strong bound 13 we obtain

$$\|\varepsilon_t\| \le \beta_2^t \|\varepsilon_0\| + 2L\sqrt{d}\epsilon \sum_{i=1}^{t} \beta_2^{t-i+1} + (1-\beta_2)\|\sum_{i=1}^{t} \beta_2^{t-i}\delta_t\|.$$

Taking expectations on both sides,

$$\mathbb{E}\|\varepsilon_t\| \le \beta_2^t \|\varepsilon_0\| + \frac{2L\sqrt{d}\varepsilon}{1-\beta_2} + (1-\beta_2)\|\sum_{i=1}^{t} \beta_2^{t-i}\delta_t\|.$$

Note that r.v.s $(\delta_i)_{1\le i \le t}$ are mean zero, using B.11, we have

$$\mathbb{E}\left\|\sum_{i=1}^{t} \beta_2^{t-i}\delta_i\right\| = \sqrt{\mathbb{E}\sum_{i=1}^{t} \beta_2^{2t-2i}\frac{\sigma^2}{N}} \le \frac{\sigma}{\sqrt{N(1-\beta_2^2)}}$$

Hence,

$$\mathbb{E}\|\varepsilon_t\| \le \beta_2^t \|\varepsilon_0\| + \frac{2L\sqrt{d}\varepsilon}{1-\beta_2} + \frac{\sigma}{\sqrt{N(1+\beta_2)}}.$$

Note that $M_0 = 0$ under our setting, so $\varepsilon_0 = -\nabla f(x_0)$, we have

$$\mathbb{E}\|\varepsilon_t\| \le \beta_2^t \|\nabla f(x_0)\| + \frac{2L\sqrt{d}\varepsilon}{1-\beta_2} + \frac{\sigma}{\sqrt{N(1+\beta_2)}}.$$

$\square$

**Lemma B.11** (Cumulative error of stochastic gradient [4]). *Assume the same settings as in Theorem B.6. Define $Y_k := \sum_{l=1}^{k} \alpha_\ell \delta_l$ where $\delta_t := \overline{g}_t - \nabla f(x_t)$ with $\overline{g}_t = \sum_{i=1}^{N} g_t^i$ and $g_t^i := \nabla f(x_t; \xi_t^i)$ following the update in (11), and $\{\alpha_\ell : \ell = 0, 1, \ldots\}$ is a deterministic sequence. Then $Y_k$ is a martingale, and*

$$\mathbb{E}\left[\left[\sum_{l=1}^{k} \alpha_l \delta_l\right]^2\right] = \frac{1}{N}\sum_{l=1}^{k} \alpha_l^2 \sigma^2.$$

*Proof.* We simply check the definition of martingales. First, we have

$$\mathbb{E}[|Y_k|] = \mathbb{E}\left[\left|\sum_{l=1}^{k}\alpha_l\delta_l\right|\right]$$

$$\leq \sum_l |\alpha_l|\mathbb{E}[|\delta_l|] \qquad \text{//triangle inequality}$$

$$= \sum_l |\alpha_l|\mathbb{E}[\mathbb{E}[|\delta_l||x_l]] \qquad \text{//law of total probability}$$

$$\leq \sum_l |\alpha_l|\mathbb{E}[\sqrt{\mathbb{E}[\delta_l^2|x_l]}] \qquad \text{//Jensen's inequality}$$

$$\leq \sum_l |\alpha_l|\sigma < \infty \qquad \text{//Assumption B.3.}$$

Second, again using the law of total probability,

$$\mathbb{E}[Y_{k+1}|Y_1,...,Y_k] = \mathbb{E}\left[\sum_{l=1}^{k+1}\alpha_l\delta_l\,\middle|\,\alpha_1\delta_1,...,\alpha_k\delta_k\right]$$

$$= Y_k + \alpha_{k+1}\mathbb{E}[\delta_{k+1}|\alpha_1\delta_1,...,\alpha_k\delta_k]$$

$$= Y_k + \alpha_{k+1}\mathbb{E}[\mathbb{E}[\delta_{k+1}|x_{k+1},\alpha_1\delta_1,...,\alpha_k\delta_k]|\alpha_1\delta_1,...,\alpha_k\delta_k]$$

$$= Y_k + \alpha_{k+1}\mathbb{E}[\mathbb{E}[\delta_{k+1}|x_{k+1}]|\alpha_1\delta_1,...,\alpha_k\delta_k]$$

$$= Y_k.$$

This completes the proof that it is a martingale. We now make use of the properties of martingale difference sequences to establish a variance bound on the martingale.

$$\mathbb{E}[[\sum_{l=1}^{k}\alpha_l\delta_l]^2] = \sum_{l=1}^{k}\mathbb{E}[\alpha_l^2\delta_l^2] + 2\sum_{l<j}\mathbb{E}[\alpha_l\alpha_j\delta_l\delta_j]$$

$$= \sum_{l=1}^{k}\alpha_l^2\mathbb{E}[\mathbb{E}[\delta_l^2|\delta_1,...,\delta_{l-1}]] + 2\sum_{l<j}\alpha_l\alpha_j\mathbb{E}\left[\delta_l\mathbb{E}[\mathbb{E}[\delta_j|\delta_1,...,\delta_{j-1}]|\delta_l]\right]$$

$$= \sum_{l=1}^{k}\alpha_l^2\mathbb{E}[\mathbb{E}[\mathbb{E}[\delta_l^2|x_l,\delta_1,...,\delta_{l-1}]|\delta_1,...,\delta_{l-1}]] + 0$$

$$= \frac{1}{N}\sum_{l=1}^{k}\alpha_l^2\sigma^2.$$

$\square$

As a direct result of Lemma B.11, we have the following.

**Lemma B.12.** *Under the same settings as in Theorem 3.6, we have*

$$\mathbb{E}\left\|\tilde{m}_{t+1}^i - \mathbb{E}[\tilde{m}_{t+1}^i]\right\|^2 \leq \left(\beta_1^2(1-\beta_2)\frac{1}{1+\beta_2} + (1-\beta_1)^2\right)\sigma^2.$$

*Proof.*

$$\tilde{m}_{t+1}^i = \beta_1 m_t^i + (1-\beta_1)g_t^i$$

$$= \beta_1(1-\beta_2)\left(g_{t-1}^i + \beta_2 g_{t-2}^i + \cdots + \beta_2^{t-1}g_0^i\right) + (1-\beta_1)g_t^i.$$

Note that

$$\beta_1^2(1-\beta_2)^2\left(1+\beta_2^2+\cdots+\beta_2^{2(t-1)}\right) + (1-\beta_1)^2 = \beta_1^2(1-\beta_2)^2\frac{1-\beta_2^{2t}}{1-\beta_2^2} + (1-\beta_1)^2.$$

By using lemma B.11, we have

$$\mathbb{E}\left\|\tilde{m}_{t+1}^i - \mathbb{E}[\tilde{m}_{t+1}^i]\right\|^2 \leq \left(\beta_1^2(1-\beta_2)\frac{1}{1+\beta_2} + (1-\beta_1)^2\right)\sigma^2.$$

$\square$

### B.2.2 Averaging Update Convergence

Assume $f\colon \mathbb{R}^d \to \mathbb{R}$ is $L$-smooth, $N$ is the number of workers, on the $i$-th worker, consider the following scheme based on the averaging:

$$
\begin{aligned}
g_t^i &:= \nabla f(x_t; \xi_t^i), \qquad \forall i = 1, \ldots, N \\
m_{t+1}^i &= \beta_2 m_t^i + (1 - \beta_2) g_t^i, \qquad \forall i = 1, \ldots, N \\
\tilde{m}_{t+1}^i &= \beta_1 m_t^i + (1 - \beta_1) g_t^i, \qquad \forall i = 1, \ldots, N \\
x_{t+1} &= x_t - \epsilon \left( \frac{1}{N} \sum_{i=1}^N \operatorname{sign}(\tilde{m}_{t+1}^i) + \lambda x_t \right). \qquad \text{//Average aggregation}
\end{aligned}
\tag{15}
$$

**Theorem B.13** (Convergence in Phase II). *Under Assumption B.2 B.3, consider the scheme in* (15)*, and $\beta_1, \beta_2 \in (0, 1)$, and $\beta_2 > \beta_1$, and $\epsilon, \lambda > 0$. $\|\lambda x_0\|_\infty \le 1$. We have*

$$
\frac{1}{T} \sum_{t=1}^T \mathbb{E}\mathcal{S}(x_t) \le \frac{f(x_0) - f^*}{T\epsilon} + \frac{2\beta_1 \beta_2 \sqrt{d}\|\nabla f(x_0)\|}{T(1 - \beta_2)} + \frac{4\beta_1 L\epsilon d}{1 - \beta_2} + \frac{2\beta_1 \sigma}{\sqrt{1 + \beta_2}} + 2(1 - \beta_1)\sigma + 2L\epsilon d.
$$

*Proof.* For notation, write $\tilde{M}_{t+1} = \sum_{i=1}^N \operatorname{sign}(\tilde{m}_{t+1}^i)$. This yields $x_{t+1} = x_t - \epsilon \tilde{M}_{t+1} - \epsilon \lambda x_t$.

Following Theorem B.1 from phase 1, once we have $\|\lambda x_0\|_\infty \le 1$, we stay within the constraint set with $\|\lambda x_t\| \le 1$ for all subsequent time $t \ge 0$.

Following a similar procedure in B.6, we have

$$
\begin{aligned}
f(x_{t+1}) - f(x_t) &\le \langle \nabla f(x_t), x_{t+1} - x_t \rangle + \frac{L}{2}\|x_{t+1} - x_t\|_2^2 \\
&\le -\epsilon \langle \nabla f(x_t), \tilde{M}_{t+1} + \lambda x_t \rangle + \frac{L}{2}\|x_{t+1} - x_t\|_2^2 \\
&\le -\epsilon \langle \nabla f(x_t), \operatorname{sign}(\nabla f(x_t)) + \lambda x_t \rangle + \frac{L}{2}\|x_{t+1} - x_t\|_2^2 \\
&\quad + \epsilon \langle \nabla f(x_t), \operatorname{sign}(\nabla f(x_t)) - \tilde{M}_{t+1} \rangle \\
&\le -\epsilon \mathcal{S}(x_t) + 2L\epsilon^2 d + \epsilon \langle \nabla f(x_t), \operatorname{sign}(\nabla f(x_t)) - \tilde{M}_{t+1} \rangle.
\end{aligned}
$$

Let us bound the last term $\langle \nabla f(x_t), \operatorname{sign}(\nabla f(x_t)) - \tilde{M}_{t+1} \rangle$,

$$
\begin{aligned}
&\mathbb{E}\langle \nabla f(x_t), \operatorname{sign}(\nabla f(x_t)) - \tilde{M}_{t+1} \rangle \\
&= \mathbb{E}\left\langle \nabla f(x_t), \operatorname{sign}(\nabla f(x_t)) - \frac{1}{N} \sum_{i=1}^N \operatorname{sign}(\tilde{m}_{t+1}^i) \right\rangle \\
&= \sum_{i=1}^N \frac{1}{N} \mathbb{E}\langle \nabla f(x_t), \operatorname{sign}(\nabla f(x_t)) - \operatorname{sign}(\tilde{m}_{t+1}^i) \rangle \\
&= \mathbb{E}\langle \nabla f(x_t), \operatorname{sign}(\nabla f(x_t)) - \operatorname{sign}(\tilde{m}_{t+1}^i) \rangle \qquad \text{//}\{\tilde{m}_{t+1}^i\}_{1 \le i \le N} \text{ are independent} \\
&\le 2\sqrt{d}\,\mathbb{E}\left\|\nabla f(x_t) - \tilde{m}_{t+1}^i\right\| \qquad \text{//Lemma B.7} \\
&\le 2\sqrt{d}\,\mathbb{E}\left[\beta_1 \left\|\nabla f(x_t) - m_t^i\right\| + (1 - \beta_1)\left\|\nabla f(x_t) - g_t^i\right\|\right] \qquad \text{//triangle inequality} \\
&\le 2\sqrt{d}\left(\beta_1 \left(\beta_2^t \|\nabla f(x_0)\| + \frac{2L\epsilon\sqrt{d}}{1 - \beta_2} + \frac{\sigma}{\sqrt{1 + \beta_2}}\right) + (1 - \beta_1)\sigma\right). \qquad \text{//Lemma B.10}
\end{aligned}
$$

Then we have

$$
\begin{aligned}
f(x_{t+1}) - f(x_t) &\le -\epsilon \mathcal{S}(x_t) + 2L\epsilon^2 d + \epsilon \langle \nabla f(x_t), \operatorname{sign}(\nabla f(x_t)) - \tilde{M}_{t+1} \rangle \\
&\le -\epsilon \mathcal{S}(x_t) + 2L\epsilon^2 d + 2\epsilon\sqrt{d}\left(\beta_1 \left(\beta_2^t \|\nabla f(x_0)\| + \frac{2L\epsilon\sqrt{d}}{1 - \beta_2} + \frac{\sigma}{\sqrt{1 + \beta_2}}\right) + (1 - \beta_1)\sigma\right).
\end{aligned}
$$

Hence, a telescope yields

$$\frac{1}{T}\sum_{t=1}^{T}\mathbb{E}\mathcal{S}(x_t) \leq \frac{f(x_0)-f^*}{T\epsilon} + \frac{2\beta_1\beta_2\sqrt{d}\|\nabla f(x_0)\|}{T(1-\beta_2)} + \frac{4\beta_1 L\epsilon d}{1-\beta_2} + \frac{2\beta_1\sigma\sqrt{d}}{\sqrt{1+\beta_2}} + 2(1-\beta_1)\sqrt{d}\sigma + 2L\epsilon d.$$

$\square$

### B.2.3 Global Lion Convergence

Assume $f\colon \mathbb{R}^d \to \mathbb{R}$ is $L$-smooth, $N$ is the number of workers, on the $i$-th worker, consider the following scheme based on the global Lion:

$$
\begin{aligned}
g_t^i &:= \nabla f(x_t; \xi_t^i) \\
m_{t+1}^i &= \beta_2 m_t^i + (1-\beta_2)g_t^i \\
\tilde{m}_{t+1}^i &= \beta_1 m_t^i + (1-\beta_1)g_t^i \\
x_{t+1} &= x_t - \epsilon\left(\text{sign}(\frac{1}{N}\sum_{i=1}^{N}\tilde{m}_{t+1}^i) + \lambda x_t\right). \qquad \textcolor{magenta}{//\text{Global Lion}}
\end{aligned}
\tag{16}
$$

**Theorem B.14** (Convergence in Phase II). *Under Assumption B.2 and B.3, consider the scheme in (16), and $\beta_1, \beta_2 \in (0,1)$, and $\beta_2 > \beta_1$, and $\epsilon, \lambda > 0$. $\|\lambda x_0\|_\infty \leq 1$. We have*

$$\frac{1}{T}\sum_{t=1}^{T}\mathbb{E}\mathcal{S}(x_t) \leq \frac{f(x_0)-f^*}{T\epsilon} + \frac{2\beta_1\beta_2\sqrt{d}\|\nabla f(x_0)\|}{T(1-\beta_2)} + \frac{4\beta_1 L\epsilon d}{1-\beta_2} + \frac{2\sqrt{d}\sigma}{\sqrt{N}}.$$

*Proof.* For notation, write $\tilde{G}_{t+1} = \frac{1}{N}\sum_{i=1}^{N}\tilde{m}_{t+1}^i$. This yields $x_{t+1} = x_t - \epsilon\,\text{sign}(\tilde{G}_{t+1}) - \epsilon\lambda x_t$.

Following Theorem B.1 from phase 1, once we have $\|\lambda x_0\|_\infty \leq 1$, we stay within the constraint set with $\|\lambda x_t\| \leq 1$ for all subsequent time $t \geq 0$.

Following the same procedure in B.6, we have

$$
\begin{aligned}
f(x_{t+1}) - f(x_t) &\leq \langle \nabla f(x_t), x_{t+1}-x_t\rangle + \frac{L}{2}\|x_{t+1}-x_t\|_2^2 \\
&\leq -\epsilon\langle \nabla f(x_t), \text{sign}(\tilde{G}_{t+1}) + \lambda x_t\rangle + \frac{L}{2}\|x_{t+1}-x_t\|_2^2 \\
&\leq -\epsilon\langle \nabla f(x_t), \text{sign}(\nabla f(x_t)) + \lambda x_t\rangle + \frac{L}{2}\|x_{t+1}-x_t\|_2^2 \\
&\quad + \epsilon\langle \nabla f(x_t), \text{sign}(\nabla f(x_t)) - \text{sign}(\tilde{G}_{t+1})\rangle \\
&\leq -\epsilon\mathcal{S}(x_t) + 2L\epsilon^2 d + \epsilon\langle \nabla f(x_t), \text{sign}(\nabla f(x_t)) - \text{sign}(\tilde{G}_{t+1})\rangle.
\end{aligned}
$$

Let us bound $\langle \nabla f(x_t), \text{sign}(\nabla f(x_t)) - \text{sign}(\tilde{G}_{t+1})\rangle$,

$$\mathbb{E}\langle \nabla f(x_t), \text{sign}(\nabla f(x_t)) - \text{sign}(\tilde{G}_{t+1})\rangle$$

$$= \mathbb{E}\langle \nabla f(x_t), \text{sign}(\nabla f(x_t)) - \text{sign}(\frac{1}{N}\sum_{i=1}^{N}\tilde{m}_{t+1}^i)\rangle$$

$$\leq 2\sqrt{d}\mathbb{E}\left\|\nabla f(x_t) - \frac{1}{N}\sum_{i=1}^{N}\tilde{m}_{t+1}^i\right\| \qquad \textcolor{magenta}{//\text{Lemma B.7}}$$

$$\leq 2\sqrt{d}\mathbb{E}\left[\beta_1\left\|\nabla f(x_t) - \frac{1}{N}\sum_{i=1}^{N}m_t^i\right\| + (1-\beta_1)\left\|\nabla f(x_t) - \frac{1}{N}\sum_{i=1}^{N}g_t^i\right\|\right] \qquad \textcolor{magenta}{//\text{triangle inequality}}$$

$$\leq 2\sqrt{d}\left(\beta_1\left(\beta_2^t\|\nabla f(x_0)\| + \frac{2L\epsilon\sqrt{d}}{1-\beta_2} + \frac{\sigma}{\sqrt{N(1+\beta_2)}}\right) + \frac{(1-\beta_1)\sigma}{\sqrt{N}}\right) \qquad \textcolor{magenta}{//\text{Lemma B.10}}$$

$$\leq 2\sqrt{d}\left(\beta_1\left(\beta_2^t\|\nabla f(x_0)\| + \frac{2L\epsilon\sqrt{d}}{1-\beta_2}\right) + \frac{(1-\beta_1)\sigma}{\sqrt{N}}\right).$$

Then we have

$$f(x_{t+1}) - f(x_t) \leq -\epsilon \mathcal{S}(x_t) + 2L\epsilon^2 d + \epsilon \langle \nabla f(x_t), \mathrm{sign}(\nabla f(x_t)) - \tilde{M}_{t+1} \rangle$$

$$\leq -\epsilon \mathcal{S}(x_t) + 2L\epsilon^2 d + 2\epsilon\sqrt{d}\left( \beta_1 \left( \beta_2^t \|\nabla f(x_0)\| + \frac{2L\epsilon\sqrt{d}}{1-\beta_2} \right) + \frac{(1-\beta_1)\sigma}{\sqrt{N}} \right).$$

Hence, a telescope yields

$$\frac{1}{T}\sum_{t=1}^{T} \mathbb{E}\mathcal{S}(x_t) \leq \frac{f(x_0) - f^*}{T\epsilon} + \frac{2\beta_1\beta_2\sqrt{d}\|\nabla f(x_0)\|}{T(1-\beta_2)} + \frac{4\beta_1 L\epsilon d}{1-\beta_2} + \frac{2(1-\beta_1)\sqrt{d}\sigma}{\sqrt{N}} + 2L\epsilon d.$$

$\square$

