# OpenReview forum: "Communication Efficient Distributed Training with Distributed Lion"
_NeurIPS.cc/2024/Conference — NeurIPS 2024 poster_

### Official Review · Reviewer_gUvC · 2024-06-26

**Soundness:** 3
**Presentation:** 3
**Contribution:** 2
**Rating:** 6
**Confidence:** 2

**Summary:**

The paper introduces Distributed Lion, a variant of the Lion optimizer, tailored for distributed training environments. Lion, known for its memory and computational efficiency, is adapted to reduce communication costs between workers and a central server. This is achieved by communicating binary or low-precision vectors rather than high-precision floating-point vectors. The paper presents theoretical convergence properties and empirical results that demonstrate Distributed Lion’s robustness and efficiency across various tasks, worker counts, and batch sizes. It shows comparable performance to standard Lion or AdamW optimizers but with significantly reduced communication bandwidth.

**Strengths:**

+ Innovation in Communication Efficiency: The use of binary or low-precision vectors for communication significantly reduces bandwidth requirements, which is a critical factor in distributed training.
+ Theoretical Validation: The paper provides a solid theoretical foundation confirming the convergence properties of Distributed Lion.
+ Empirical Evidence: Extensive experiments demonstrate the robustness and efficiency of Distributed Lion across a variety of tasks, making a strong case for its practical applicability.

**Weaknesses:**

- Incompatible with Allreduce: after converting the gradients to binary or low-precision, Allreduce cannot be used for gradient synchronization. One of my concerns is about its communication efficiency in real-world distributed systems, especially training with a high number of workers.
- Computation Overhead: While the communication cost is reduced, the overhead of converting updates to binary or low-precision vectors and back might offset some of the gains in certain scenarios. It helps if the end-to-end training throughput comparison is reported.

**Questions:**

What is the fundamental difference between distributed Lion and SIGNUM-like algorithms?

---

> ### Author Rebuttal · Authors · 2024-08-07
>
> **1. Incompatible with all reduce.**
>
> Our current algorithm indeed requires a customized all_reduce, but we believe the code should be relatively simple to apply to various real-world applications. Additionally, we are exploring ways to optimize the communication process for low-bit information.
> You are right, due to the limited operator support in practical all reduce such as PyTorch, which is ring-based reduce, it makes the low-bit information hard to broadcast and maintain its information when broadcast through the whole ring. This issue widely exists in all our baselines. Our current practical solution is packing and encoding our binary information into a low-bit tensor and then using an int8 tensor as a container to broadcast through the ring all reduce. This can still compress the information from 2-4x depending on the number of workers, we show the empirical result under different works below together with question 2.
> For sure, we are looking forward to some core updates that can be made by the NCCL library or pytorch team to support more low-bit friendly operators or datatypes like fp4, and fp8 during communication, which could greatly help with low-bit communication algorithms.
>
> **2. Computation overhead.**
>
> Please check the common response.
>
> **3. The difference between Distributed Lion and SIGNUM-like algorithm.**
>
> The main differences are two-fold:
> - Lion has an additional reweighting before the sign(), as in sign( $\beta_1 m_t + (1 - \beta_1) g_t$) and the weight decay term, which are crucial for performance improvement. The unique form of Lion presents the special $\ell_\infty$ constraint optimization view of the algorithm.
>
> - Usually, when applying SIGNUM to distributed training, we aggregate the gradient and then apply the sign operation. In distributed Lion, we transmit the signed results and their majority vote, not the gradient. The weight decay, as a result, is applied locally in distributed Lion.

---

### Official Review · Reviewer_aXYH · 2024-06-28

**Soundness:** 2
**Presentation:** 3
**Contribution:** 2
**Rating:** 5
**Confidence:** 5

**Summary:**

Large-scale AI model training has increasingly higher requirements on time, cost and environmental impact, so it is crucial to develop efficient optimizers. As an emerging optimizer, Lion optimizer has advantages in memory, computation and sample efficiency compared with AdamW. Distributed Lion: The paper proposes Distributed Lion, which is an innovative adaptation of Lion optimizer in distributed training environment. Using symbolic operations in Lion, Distributed Lion only requires binary or low-precision vectors to be communicated between working nodes and central servers, significantly reducing communication costs.

**Strengths:**

1. Distributed Lion significantly reduces communication overhead by communicating only binary or low-precision vectors between workers, which is particularly beneficial for large-scale distributed training.
2. The paper provides theoretical analysis to prove the convergence of Distributed Lion.
3. Experimental results show that Distributed Lion can achieve comparable performance to the standard Lion or AdamW optimizer while reducing communication bandwidth.

**Weaknesses:**

1. The actual updating on local worker parameters is gradients, while the communicated message is signs. While the theoretical analysis shows this updating can guarantee the convergence, the actual updating style looks like the quantization. The important baselines like QSGD and SignSGD are missed.
2. The performance of Distributed Lion can be sensitive to hyperparameter choices, especially those related to communication and aggregation strategies.
3. The code is not provided. Thus the reproducibility of the experiments is weakened.
4. The experiment performance on the CIFAR-10 is very low. Considering that the well-known validation performance of CIFAR-10 can be achieved as 94%, the proposed results are around 90%. Why the performance decreases?
5. The important baseline SGD with momentum is not provided.
6. The convergence curves on training with ImageNet and OpenWebText are not provided. This makes it hard to identify the convergence speedup between different optimizers.
7. The wall-clock time is not provided. The quantization operation and the majority vote require extra time, it will be better to show this optimizer can reduce the real-world throughputs.

**Questions:**

1. How do you ensure that the hyper-parameters of different optimizers are set as a suitable combination for them? Suitable hyper-parameter settings can ensure the fair comparison.

If the above weaknesses and questions are addressed, I'm happy to raise the score.

**Limitations:**

Two minor points considering the realistic device constraints:
1. The experiment scalability is with 32 GPUs, which is not a large scale distributed training setting.
2. The training model is small-scale with less than 1B parameters.

---

> ### Author Rebuttal · Authors · 2024-08-07
>
> **1. Comparison to quantization methods.**
>
> The actual update on each worker is actually not the gradient, but rather the Lion’s update (the sign() plus weight decay). To our knowledge, the quantization methods often quantize the gradients before feeding the quantized gradient to the optimizer. In our case, the sign() operation is applied to the reweighted momentum, which is computed using exact gradients. Moreover, unlike quantization methods, the sign() function in distributed Lion is a feature, instead of any approximation.
>
> Given the limited time for rebuttal, we are not able to run QSGD and SignSGD distributedly. However, note that the Ternary gradient baseline is a specific quantization method, where the gradient is quantized to a ternary. Moreover, we provide a comparison against a stronger baseline than SignSGD (Sign SGD momentum, a.k.a., SIGNUM), and also the SGD momentum (as requested by the reviewer) in the following:
>
> | Algorithm        | worker = 4 | worker = 8 | worker = 16 |
> |------------------|------------|------------|-------------|
> | SGD Mom          | 79.3       | 81.8       | 82.5        |
> | SIGNUM           | 88.3       | 87.2       | 85.5        |
> | Dist Lion (MaVo) | 90.9      | 89.6      | 88.9       |
> | Dist Lion (Avg)  | 90.5      | 87.9      | 87.9       |
>
> From the result, we see that the distributed Lion still results in better performance.
>
>
> **2. Sensitivity to hyperparameters.**
>
> Empirically we observe that the performance of distributed Lion is not sensitive to hyperparameters. In our experiments, we did not tune the betas, and chose the default betas suggested in the Lion paper, and only tuned the learning rate and weight decay, which are the two hyperparameters shared by all optimizers. **According to our findings in Table 4, the best hyperparameters for distributed Lion are the same as in the Global Lion.** Therefore, we expect that the default hyperparameters for Lion will always be a good configuration for distributed Lion.
>
> **3. Reproducibility.**
>
> The actual implementation requires further internal review to be made publicly available. But we provide an implementation of our distributed Lion optimizer in this [link](https://anonymous.4open.science/r/dist_lion-6789/dist_lion.py).
>
>
> **4. Performance on CIFAR-10.**
>
> Note that as mentioned in Line 224, we are applying all optimizers to ViT models with 6 layers and 8 heads, as it is easier and faster to train. We chose ViT models as they are broadly used in practice. As a result, given the limited size of the ViT model, a performance with nearly 90% accuracy is already pretty high. The overall architecture is similar to the ViT small from this codebase [https://github.com/kentaroy47/vision-transformers-cifar10/tree/main] (note that it is ViT small, not the ViT small (timm transfer)). The training result from their repo is around 80% accuracy.
>
> **5. Convergence Curve.**
>
> We provide the training curves of Distributed Lion, Global Lion, and Global AdamW on ImageNet training in the uploaded PDF from the common response. From the curve, we can tell that Distributed Lion performs slightly better throughout the training.
>
>
> **6. Wall-clock time comparison.**
>
> We refer the reviewer to the common response for this question.

---

> > ### Comment · Reviewer_aXYH · 2024-08-09
> > **Thanks for the responses**
> >
> > I appreciate for the efforts from authors. I have two follow-up questions:
> > - 1. Why does SIGNUM outperform SGD Mom? As a compression method, SGD Mom might be the upper bound of SIGNUM? Maybe the hyper-parameters of the SGD Mom are not well tuned?
> > - 2. Could you give some explanations about why the Optimizer State Communication (ms) of MaVo is half less than the baseline. Because the communication messages are signs, I suppose they should be less than original messages for 16x or 32x.

---

> ### Author Response · Authors · 2024-08-09
> **Reply to Reviewer**
>
> We thank the reviewer for your followups :)
>
> **1. Why SIGNUM outperforms SGD Mom.**
> This is in fact expected. So the sign() operator is not merely compressing the momentum information, it also **normalizes** the update. Take the Adam optimizer as an example, at the first step, Adam's update recovers the normalized gradient descent $ g / ||g|| $, and at each step, Adam's update is also roughly doing that. It is known that these kinds of **normalized gradient descent** perform well in practice. One reason is that it makes each entry of the parameter update at a similar pace (therefore similar to a second-order method), the other reason is that it can avoid saddle points (as the update norm is relatively constant). SIGNUM (and similarly Lion) also mimics what Adam does by having the sign operation (their update norm is a constant). Based on my personal experience, SGD Momentum works better for convolution-based networks. But for Transformer models, often Adam-like optimizers work better.
>
> **2. Why the Optimizer State Communication (ms) is half less?**
> As we mentioned in the common response, currently during our implementation, we are using the int8 all reduce instead of binary all reduce. This is because in practice all-reduce in PyTorch does not support low-bit all-reduce in a ring-based fashion, which is the default way to perform distributed training. Our current practical solution is packing and encoding our binary information into a low-bit tensor and then using an int8 tensor as a container to broadcast through the ring all reduce. This can still achieve a 2-4x compression, depending on the number of workers.
>
> On the other hand, we are looking forward to some core updates from the NCCL library or pytorch team to support more low-bit friendly operators or datatypes like fp4, and fp8 during communication, which could greatly help with low-bit communication algorithms. As a result, our current work mainly focuses on demonstrating the theoretical convergence and empirical performance of distributed Lion algorithms, rather than the practical aspects of implementation.

---

> > ### Comment · Reviewer_aXYH · 2024-08-13
> > **Thanks for the further explanations**
> >
> > I appreciate the further explanations, which clearly address my concerns. I'd like to increase my score as 5.

---

### Official Review · Reviewer_ZPCr · 2024-07-12

**Soundness:** 3
**Presentation:** 3
**Contribution:** 2
**Rating:** 6
**Confidence:** 4

**Summary:**

This paper extends the Lion optimizer to data parallel distributed training. Unlike optimizers like SGD and Adam, the binary update in Lion can be exploited to minimize the communication. They investigate two cost effective methods for the communication of  binary updates; averaging and majority vote. Experimental results show that both methods yield competitive results to global Lion and AdamW.

**Strengths:**

The convergence analysis provided in Section 3 gives some reassurance to this non-conventional optimization method. Results are promising and experimental conditions seem adequate.

**Weaknesses:**

The proposed method is a trivial extension of Lion to data parallel distributed training, so the only interesting contribution seems to be the convergence analysis.

The main contribution of this work is supposed to be the reduction of communication overhead, but there are no results showing the actual breakdown of the training time. Therefore, it is not possible to determine whether the reduction of communication volume is actually contributing to the reduction of the overall training time. Since the results seem to vary quite a bit for different models and datasets, such information is useful for determining whether the experiments are conducted for configurations that actually show a significant impact on the training time. There remains a possibility that the current method does not work as well for extremely large models trained with ZeRO 3 data parallelism, which is where the communication overhead really becomes a problem.

**Questions:**

How different are the global binary updates between averaging and majority vote?
Are the results similar because they are similar or despite their large difference?

**Limitations:**

The limitations pointed out above are not explicitly stated in the paper.

---

> ### Author Rebuttal · Authors · 2024-08-07
>
> **1. Wall-clock time reduction.**
>
> We refer the reviewer to the common response for this question.
>
>
> **2. Compatibility with ZeRO3 data parallelism.**
>
> Although large-scale parallelism techniques such as ZeRO3 and FSDP require additional inter-node gather operations that cannot be accelerated by our algorithm, we can still optimize intra-node optimizer state synchronization. In the common large-scale training scenario where the model size increases and intra-node latency becomes significant, our optimizer can reduce communication time by around 50%. While our algorithm does not reduce inter-node gather time in this case, it is compatible with other gradient-size compression algorithms like Galore[1]. In optimal cases, combining these algorithms can lead to even greater reductions in communication time.
>
> **3. Difference between majority vote and averaging.**
>
> Based on our empirical observations, we do not observe too much difference between the two schemes and one could potentially perform better than the other on different datasets. The two reduction schemes are indeed similar, as the majority vote method can be seen as applying an actual sign() activation on top of the averaging.
>
> ---
>
> [1] Zhao et al, GaLore: Memory-Efficient LLM Training by Gradient Low-Rank Projection.

---

> > ### Comment · Reviewer_ZPCr · 2024-08-13
> >
> > Thank you for providing the actual training time and its breakdown. I still feel that more emphasis should be put on the reduction of communication time for different model sizes, batch sizes, and including other modes of parallelism such as tensor, pipeline, and ZeRO. The communication in Int8 does reduce the volume by half, but this method has the potential to do much better. The effort to extend Lion to a distributed Lion using only data parallelism is minimal, so I feel that the paper should include more detailed studies on the actual reduction of communication time. I am sticking to my original score.

---

### Official Review · Reviewer_7ynU · 2024-07-12

**Soundness:** 3
**Presentation:** 3
**Contribution:** 3
**Rating:** 7
**Confidence:** 3

**Summary:**

This paper proposes Distributed Lion, a new variant of Lion optimizer for distributed training. The proposed algorithm only requires to communicate binary or lower-precision vectors between workers to the center server, significantly reducing the communication cost. The theoretical analysis proves the convergence of the proposed algorithms. The empirical results show that the proposed algorithms have comparable model performance on CV/NLP applications but with significantly less communication overhead compared to the baselines.

**Strengths:**

1. This paper proposes Distributed Lion, a new variant of Lion optimizer for distributed training.

2. The proposed algorithm only requires to communicate binary or lower-precision vectors between workers to the center server, significantly reducing the communication cost.

3. The theoretical analysis proves the convergence of the proposed algorithms. The empirical results show that the proposed algorithms have comparable model performance on CV/NLP applications but with significantly less communication overhead compared to the baselines.

**Weaknesses:**

1. According to Assumption 3.1, the convergence requires i.i.d. local datasets, while real-world distributed training typically uses non-i.i.d. local data.

2. In the empirical results, there seems to be no wall-clock time for training is reported. Note that the overall goal of communication reduction is to reduce the training time. Thus, it is important to report loss/acc vs. wall-clock time in the experiments.

**Questions:**

1. Is it possible to provide a convergence analysis based on non-i.i.d. data?

2. For the experiments, are the local dataset on each worker i.i.d. or non-i.i.d.?

3. Since the proposed algorithm can compress the communication to an extreme extend, I wonder whether it could also be applied to the federated learning scenario, where the local datasets are not only non-i.i.d., but also highly heterogeneous.

4. Is there any empirical results reporting wall-clock time of training?

**Limitations:**

The limitations are well discussed and addressed in this paper.

---

> ### Author Rebuttal · Authors · 2024-08-07
>
> **1. i.i.d assumption.**
>
> Indeed, currently, we assume data are i.i.d (the dataset on each worker is pre-sharded before training). We leave it as a future work to show the convergence of distributed Lion under a non-i.i.d setting.
>
>
> **2. Wall-clock time comparison and communication reduction.**
>
> Please check the common response.
>
>
> **3. Whether the method can be applied to federated learning?**
>
> Yes, in principle we expect the method to work in a federated fashion. However, when distributed Lion is applied in a local-SGD fashion, the local update will involve multiple steps of the Lion update, so the aggregated server update will be quantized instead of being binary.

---

> > ### Comment · Reviewer_7ynU · 2024-08-12
> >
> > My major concern on wall-clock time comparison is well addressed. And I guess the non-iid settings and the federated learning settings worth another paper in the future.
> >
> > Thus, I will keep the positive score.

---

### Author Rebuttal · Authors · 2024-08-07

We sincerely thank the reviewers for their valuable feedback. In the following, we provide the general response and address individual concerns separately in individual responses.

**1. Wall-clock time comparison.**

Several reviewers have requested a wall-clock time comparison. Our study primarily focuses on the theoretical and empirical performance of distributed Lion algorithms, rather than the practical aspects of implementation. Currently, we employ a majority vote strategy using int8 combined with a built-in all-reduce operation, which already achieves notable speedup in wall-clock time and reduces communication overhead. In particular, using a 3B Transformer model, with 1K token length, across 64 A100 40G devices, with batch size 1, using our current implementation, we have:

| Method                  | Forward Time (ms) | Backward Time (ms) | Optimizer State Communication (ms) | Total (ms) |
|-------------------------|-------------------|--------------------|------------------------------------|-------|
| AdamW                   | 18                | 141                | 132                                | 291   |
| Lion                    | 18                | 137                | 133                                | 288   |
| Distributed Lion (MaVo) | 18                | 128                | 61                                 | 207   |

As we can see, the communication overhead is saved by more than 50% even with our int8 implementation. This could be further improved if customized all reduce is supported for binary inputs.

Reviewer gUvC raised concerns about the limited support for low-bit operations in ring-based all-reduce frameworks like PyTorch. This challenge affects not only our approach but also all our baseline comparisons. We address this by packing binary information into low-bit tensors and then broadcasting it using an int8 tensor container through the ring all-reduce.
We anticipate future enhancements from the NCCL library or the PyTorch team that introduce more efficient low-bit operations or data types, such as fp4 or fp8. Such developments would significantly improve the efficiency of low-bit communication algorithms.

**2. Release of the Code.**

We release our implementation of the distributed Lion [here](https://anonymous.4open.science/r/dist_lion-6789/dist_lion.py). We intend to publicly release the method for the convenience of future research.

---

### Decision · Program_Chairs · 2024-09-25

**Decision:**

Accept (poster)

**Comment:**

This paper considered a distributed version of the Lion optimizer, which reduces the communication cost. The authors provided theoretical analysis of this algorithm and showed empirical results on vision and language tasks. The reviewers unanimously recommended to accept this paper. I suggest the author incoporate the extra wall-clock time experiments and communication reduction experiments in the final version of the paper.